# Formation, breaching and flood consequences of a landslide dam near Bujumbura, Burundi

Léonidas Nibigira[1], Hans-Balder Havenith[1], Pierre Archambeau[2], and Benjamin Dewals[2]

[1]Geohazards and Environment, Department of Geology, University of Liege, 4000 Liege-Belgium

[2]Hydraulics in Environmental and Civil Engineering (HECE), Research unit Urban & Environmental Engineering, University of Liege, 4000 Liege-Belgium

Correspondence to: Léonidas Nibigira (leonidas.nibigira@doct.ulg.ac.be)

**Abstract.** This paper investigates the possible formation of a landslide dam on the Kanyosha River near Bujumbura, the capital of Burundi, as well as the interplay between the breaching of this landslide dam and the flooding along the river. We present
an end-to-end analysis, ranging from the origin of the landslide up to the computation of flood waves induced by the dam breaching. The study includes three main steps. First, the mass movement site was investigated with various geophysical methods that allowed us to build a general 3D model and detailed 2D sections of the landslide. Second, this model was used for dynamic landslide process modelling with the Universal Distinct Element Code. The results showed that a fifteen-meter-high landslide dam may form on the river. Finally, a 2D hydraulic model was setup to find out the consequences of the
breaching of the landslide dam on flooding along the river, especially in an urban area located downstream. Based on 2D maps of maximum water depth, flow velocity and wave propagation time, the results highlight that neglecting the influence of such landslide dams leads to substantial underestimation of flood intensity in the downstream area.

**Keywords** Bujumbura, landslide dam, dam breaching, geomechanic and hydraulic modelling, flood propagation, multi-hazard

## 1 Introduction

The city of Bujumbura, the capital of Burundi, faces serious problems related to natural hazards. Floods are the most important natural challenge in terms of induced losses. This is aggravated by heavy tropical rains. It also becomes clear that geohazards strongly contribute to the risk of flooding. In February 2014, floods resulting from a failure of a temporarily created landslide dam caused 64 casualties. Over 940 houses were destroyed and this resulted in over 12,500 homeless people (UNITAR/UNOSAT 2014, Reliefweb 2014). This indicates that a complete assessment of flood risk should take into account
landslides which may be considered as some of the most important natural hazards in the region. They interact with the hydrographic network by forming natural dams. The formation of landslide dams is caused by the combination of several factors. Many spectacular cases were reported, in which earthquakes were as a major trigger (Adams, 1981; Cui et al., 2012). For example, the Wenchuan earthquake in 2008 caused up to 828 landslide dams (Fan et al., 2012; Fan et al., 2017). In addition to earthquakes, long and heavy rainfalls (Li et al., 2011) as well as other local parameters can lead to slope instability and to
landslide dam formation. Losses related to natural dams can occur both during and after the formation of the dam. Losses that occurred during the formation are exemplified by the cases of the village of Hsiaolin that had been entirely buried in 2011 under a massive debris flow and landslide in southern Taiwan (Li et al., 2011) or by the sweeping of Attabad and Sarat villages in Northern Pakistan in 2010 (Butt et al., 2013). In many other cases, losses are mainly linked to the dam failure and associated downstream floods. Related studies (Cui et al., 2006; Wells et al., 2007; Downs et al., 2009; Wang et al, 2016; Costa and
Schuster, 1988; Li et al., 2002; Chen et al., 2004) show that the effects of dam failure can be many times greater than those caused by the sliding during the formation of the dam. Although different methods have been proposed and applied to understand their formation and/or breaching mechanisms (Korup, 2004; Corominas and Moya, 2008; Crosta and Clague, 2009;

Dong et al., 2009; Nandi and Shakoor, 2009; Shrestha, B. and Nakagawa, 2016), each case of natural dam has its own specificities related to the local context. Therefore, case studies are very important. Unfortunately, there is a lack of both case
studies and data required for the analyses, especially in Africa. Consequently, statistical studies based on past events are missing and that is a challenge when the risk of dam formation or the breaching of an existing dam has to be assessed. This underlines the importance of scenario simulations supported by the use of modern modeling tools. In Central Africa (including Burundi), despite existing studies in the field of environmental hazard analysis (Ilunga, 2006; Moeyersons et al., 2010; Nibigira et al., 2015; Michellier et al., 2016; Jacobs et al., 2016), quantified landslide multi-hazard scenario analyses are still rare. This
lack of multi-hazard studies in equatorial Africa was highlighted recently by Jacobs et al. (2016). For the city of Bujumbura, there is a need to develop a multi-risk study, analyzing, on one hand, the hazard related to landslide activation and natural dam formation, and, on the other hand assessing the potential impacts of the dam failure on the hydrographic network.

We performed such a study to the existing mass movement called 'Banana Tree Landslide' (called BTL below). This landslide was selected for its size (it is one of the largest active landslides in the vicinity of Bujumbura with a volume of more than
$4 \times 10^6$ m$^3$) and due its position along the Kanyosha River, upstream of the city (Fig. 1a and Fig. 1b) making it a potential danger for people and infrastructures in the area.

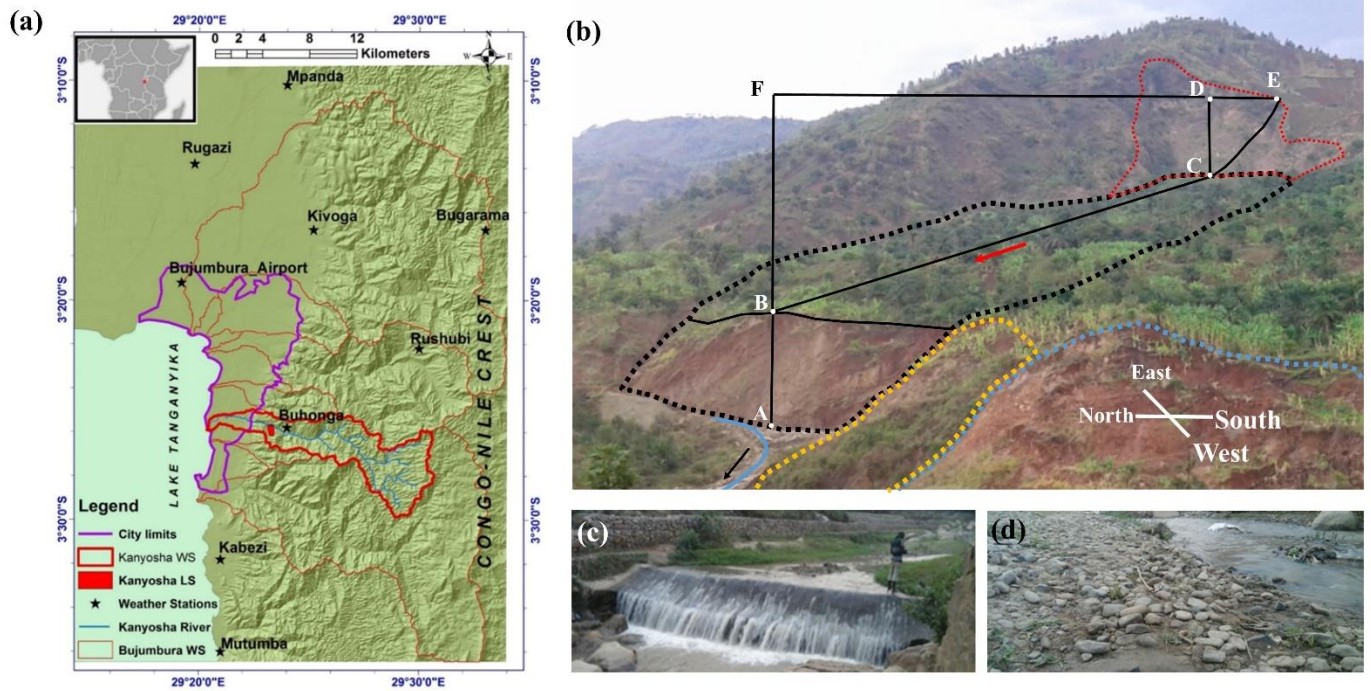

**Figure 1.** (a) Bujumbura region with indication of watersheds of the main rivers, the limits of the city and Lake Tanganyika. The watershed
of the Kanyosha River is highlighted in the central part, with the river network inside. The weather stations in and around Bujumbura, the Kanyosha Landslide (in the text called 'Banana Tree Landslide', also referred to as BTL, in red contours) and the Congo- Nile crest are also shown. 'LS' and 'WS' stand for 'Landslide' and 'Watershed', respectively. (b) View of BTL (black dotted contour) and the main scarp (red dotted contour) as well as its lateral local instabilities (orange and light blue dotted contours). The landslide sliding direction and the river flow direction are indicated by the red and the black arrows, respectively. AB indicates the height (=26 m) of the landslide frontal part near the river; BC outlines the BTL length in the sliding direction (~750 m); CD shows the height of the main scarp (~ 75m) along profile BCE.
The blue line indicates the river channel axis. (c) Waterfall over a former flood control structure. (located 270 m downstream of cross-section 3 shown in Fig. 3) (d) View of the river bed during the dry season with presence of cobbles and fine boulders that are deposited after floods during the wet season.

Since the gorge of the valley is relatively narrow in the landslide area, a displacement of the BTL of a few tens of meters would be enough to form a natural dam and a reservoir lake, which could later break with all the risks that such an event represents for the part of the city located downstream. The lifespan of natural dams cannot be known accurately and can be relatively short: it is less than one hour for 34% of the known cases investigated by Peng and Zang (2012) and 27% of all cases according to Costa and Schuster (1988). Moreover, considering the tropical climate context of the target area, it can be assumed that the reservoir behind a new dam can be quickly filled after very intense rainfalls that occur on a regular basis during the wet season. All those parameters reduce considerably the time between the dam formation and the possible dam breaching, highlighting the necessity to know in advance the consequences corresponding to different scenarios, particularly for such areas where warning systems are not very effective or just missing.

Our recent observations show that the western part of the landslide (in the foreground of Fig. 1b), with relatively soft slopes, is marked by very local slope instabilities (yellow and light blue dotted contours) that do not contribute to the general movement. However, the eastern part (black dot outlines in Fig. 1b) presents steep slopes near the river; this active zone is 250 m wide and could soon move to form a landslide dam. The presence of water ponds in this eastern part (Fig. 2a) is likely to contribute to future instability that could develop along the main sliding axis BC (shown in Fig. 1b).

In order to understand the landslide mechanisms in terms of triggering factors, evolution and effects, numerical modelling has been carried out to analyze its stability, also under dynamic (seismic) conditions. The effects of the dam and its breaching on the flood potential along the river and the consequences especially downstream in the urban area were studied through an additional hydraulic model. Simulated flood scenarios are discussed with respect to parameters such as the water depth, the flow velocity and the floodplain delineation.

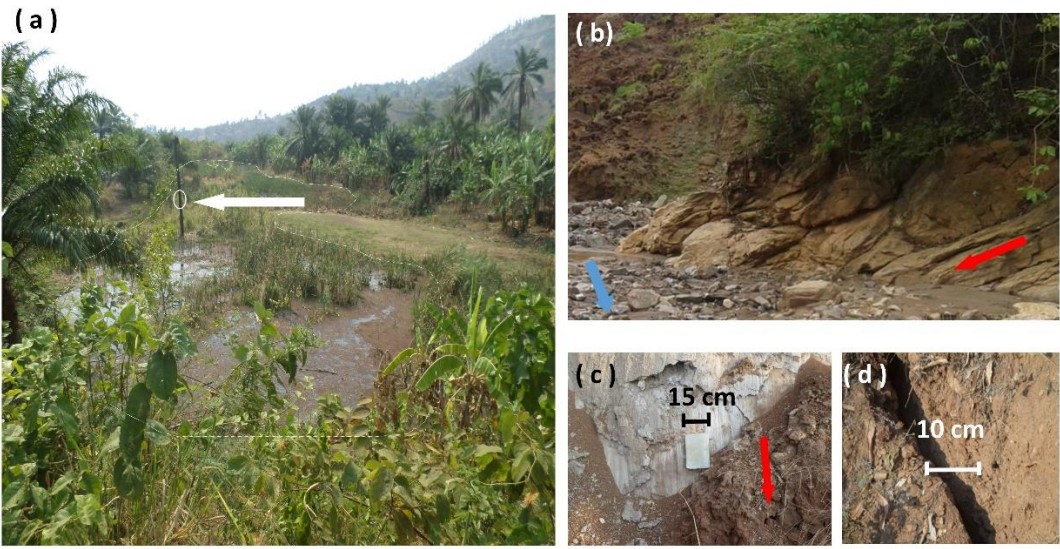

**Figure 2.** Field observations highlighting the critical stability state of BTL: (a) Pond on the landslide with an oil palm designated by the white arrow. This shows that these ponds are recent (oil palm trees do not grow in water; its particular foliage compared to others shows that its growth was stopped recently). (b) View of the rock structures at the foot of the landslide, generally dipping towards the north (left side), parallel to the sliding direction (red arrow). The blue arrow indicates the river flow direction. (c) View of a crack on the sliding interface in a clay layer. The red arrow shows the direction of sliding of the right part along the clay layer. (d) A crack found on the landslide surface.

## 2   Data and methods

### 2.1   Channel description

The Kanyosha River is one of the most important rivers in Bujumbura. Most of its watershed is located in the Congo-Nile crest, in the east side of the city. The upstream parts consist of V-shaped valley while the north and south flanks are made up of wooded areas and steep agricultural areas subject to the erosive action of the runoff descending the shoulders of the rift. The grain size of the river bed deposits is variable. Based on the extended Udden-Wentworth grain-size scale nomenclature (Terry and Goff, 2014), the riverbed material can be classified into three main groups.

- The first consists of cobbles of around 10 cm in diameter or more (Fig. 1d). The coarse part of this category consists of fine boulders, with a diameter generally under 40 cm.
- The second group is made up of isolated medium boulders that are often prone to the action of humans, carving them into building materials (mainly paving plates). This category is difficult to take into account due to its strong irregularity.
- The third group consists of silt and clay zones, generally near former hydraulic structures in the downstream part of the river. In this category, we can mention small herbaceous islets, often located near the river overbanks. As in the second group, this category is found only in small isolated and scattered areas, subject to strong seasonal variations.

Details on the Udden-Wentworth grain-size scale nomenclature are provided in Supplement 1 (Table S1). Globally, the first group is hydraulically predominant. Here, the variability of the grain size was accounted for by means of sensitivity analysis (Section 3.3). In 2006, hydraulic structures were constructed to regulate the river; but they were quickly damaged by floods during the following rainy seasons. Nonetheless, isolated coarse materials resulting from the destruction of these structures are observed. They join the second group described above. The accumulation of material upstream of the remains of the structures often form horizontal platforms, generating small waterfalls (Fig. 1c).

### 2.2   Topographic and geophysical data

We used a 10 m–resolution Digital Elevation Model (DEM) of the river valley, provided in the coordinate system UTM35S and in raster format (Fig. 3). It was produced in 2012 by the '*Bureau de Centralisation Geomatique du Burundi*'. The DEM was resampled at a resolution of 2 m × 2 m, which is the resolution used for hydraulic modelling. For the second part of the analysis, the geometry of the dam was incorporated, taking into account the results provided by the first part related to the landslide process analysis.

Given that no data were available for defining the river bathymetry and the overbank topography, the flow was computed based on the DEM. The average width of the river is about 20 m for a discharge of 3 m³/s, 32 m for 60 m³/s (20-year flood) and 40 m for 120 m³/s (50-year flood). Hence, a computational spacing of 2 m (obtained after resampling) is certainly fine enough to represent the flow field over the width of the river, since the number of computational cells over the width of the river is in-between 10 and 20.

While resampling the DEM is important for computational reasons, only the topographic details already present in the initial DEM (10 m × 10 m) are captured. Ideally, the hydraulic analysis should use a higher resolution DEM such as light detection and ranging (LiDAR) elevation data. However, in the data-scarce environment of the study area which is a commonplace in many parts of Africa, a 10 m resolution is among the best in the region, especially when compared to SRTM and ASTER GDEM provided by USGS. The example of some recent works (Jacobs et al., 2016; Alvarez et al., 2017) showed that using medium- or low-resolution products remains a valuable intermediate step to advance the understanding of flood risk in data-scarce areas in Africa, provided that the results are interpreted in the light of the uncertainties affecting the input data.

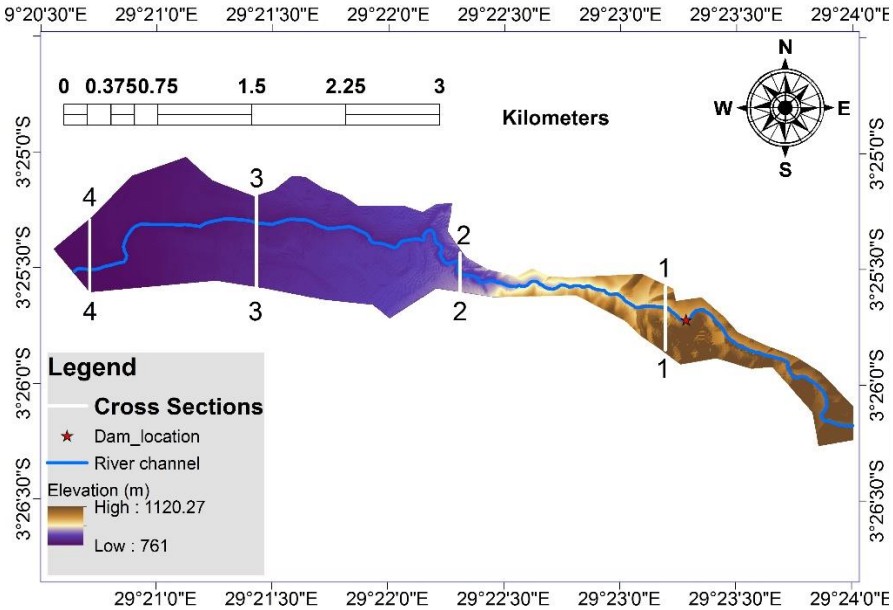

**Figure 3.** Digital elevation model (m) used for hydraulic modelling within the computational domain, with cross-sections where hydrographs were extracted and dam location. The river main channel is also highlighted.

Moreover, to assess the DEM used for hydraulic modelling, we also considered a field survey that was conducted in the study area during the dry seasons (June-September) in 2014 and in 2015. The field measurements covered the main riverbed and part of the floodplains (band of 10-20 m) of Kanyosha River, from 500 m upstream of the dam down to Lake Tanganyika. As shown in Fig. 4, the differences between the DEM used in our hydraulic simulations and data from the field survey remain moderate, as they range mostly between – 0.5 m and + 0.5 m. The median and mean differences are both - 7 cm. The RMS error between the 10 m × 10 m DEM and field measurements is 65 cm, which seems reasonable. Most significant differences are obtained near the river banks. This may result from discretization errors and/or from the instability of the banks due to planform evolution of the riverbed over the period from 2012 (when the 10 m × 10 m DEM was produced) to 2014 (field survey in the main riverbed).

In the upper part of the valley, showing a distinctive V-shape with relatively steep lateral slopes, the flow tends to concentrate in the central main channel and its vicinity. Therefore, the hydraulic modelling results should be less affected by small inaccuracies in the DEM than further downstream. A sensitivity analysis of the simulation results with respect to the inaccuracy in the topographic data is presented in Section 4.3.2.

For the landslide stability analysis, the surface data provided by the DEM were combined with subsurface information obtained by local geophysical field measurements completed in summer 2013. They consist in electrical resistivity tomography (ERT) and ambient noise HV measurements. Fig. 5 provides an overview of the measurements and two examples of ERT profiles. From these investigations, the thickness of the landslide mass and some of its geophysical properties (notably, the elastic properties) could be determined.

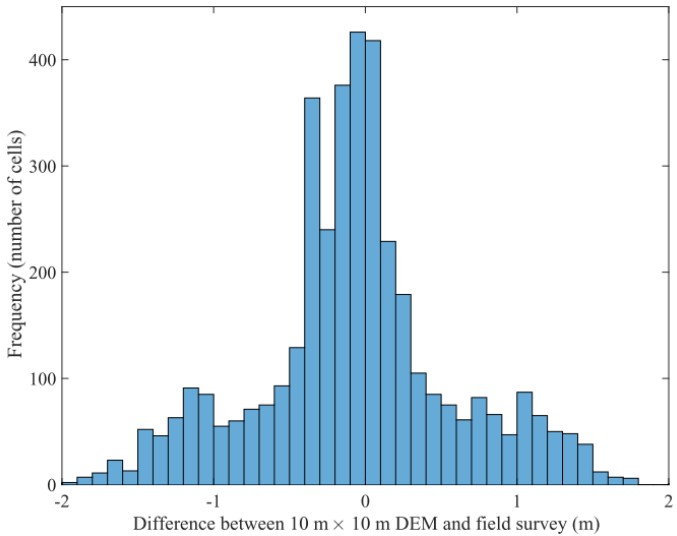

**Figure 4.** Elevation difference between the topography from field measurement and the resampled 2 m × 2 m DEM used for hydraulic modelling.

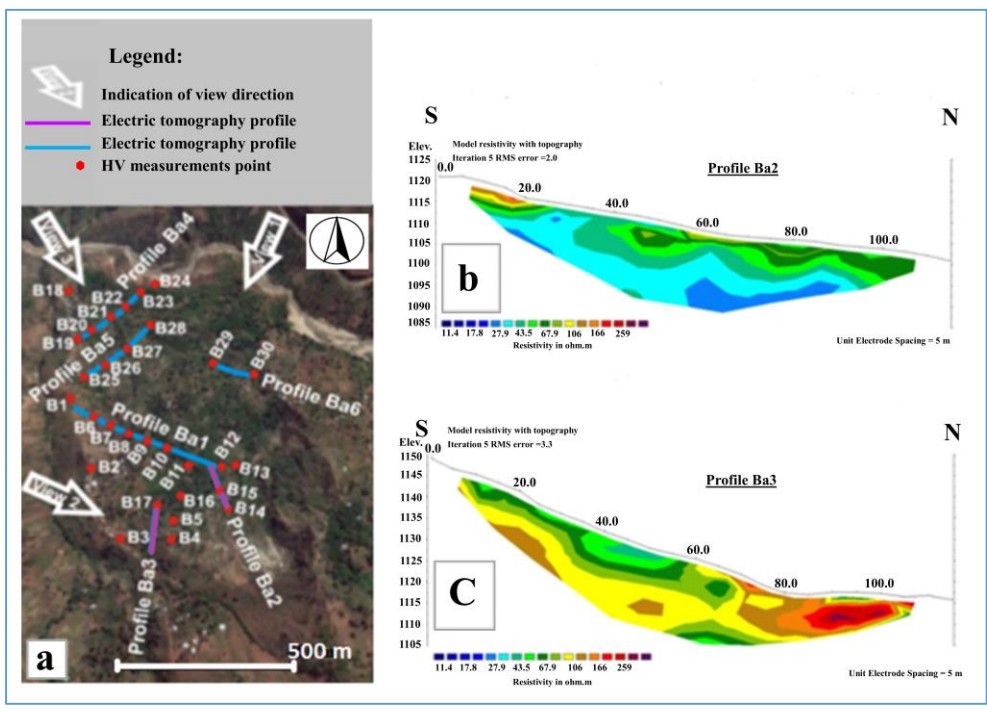

**Figure 5.** The 'Banana Tree Landslide' field measurements: overview of ERT and HV measurement locations (a), ERT profiles Ba2 (b) and Ba3 (c). In Fig. 5a, Ba2 and Ba3 profiles are highlighted in purple.

### 2.3 Landslide analysis with UDEC: model construction

From the field measurements, a model of the landslide was established. Fig. 6 corresponds to a 2D section along the main axis
of the mass movement and shows the present (actual) topography of the landslide (plain line) and the reconstructed (estimated) initial topography (before first instabilities appeared, marked by a dashed line) as well as the main sliding surface (dotted line). The initial situation is characterized by an average slope of about 15° while the current profile (red line) is marked by a clear scarp in the upper part, below which the landslide material has a thickness of about 15 m and by more massive landslide deposits (thickness of about 50 m) in the middle and the lower part towards the river.


**Figure 6.** Initial and present BTL profiles. The larger thickness of the present profile in the downstream part of the model is a result of the relative lift-up after a trans-rotational sliding and material accumulation from the upper parts of the initial profile.

On the basis of this cross-section of the landslide a slope stability analysis (both a back-analysis starting from the reconstructed pre-landslide model and a 'predictive' analysis starting from the present-day situation) as well as mass movement modelling were carried out in 2D using the Universal Distinct Element Code (UDEC). UDEC was developed by Cundall (1971) to evaluate the response of materials (discretized as blocks) to a given loading in static and dynamic (e.g. seismic) conditions. The distinct element method has been used in various studies and it is particularly suitable for rock slope stability analyses
(Kveldsvik et al., 2009, Kainthola et al., 2012, Bhasin and Kaynia 2004, Esaki et al., 1999, Chuhan et al., 1997).

For the modelling with UDEC, the landslide was subdivided into three main blocks (see numerical measurement points 12, 13, 15, in Fig. 7, which are located on the upper, middle and lower block, respectively). Cracks (joints) included between the blocks (that represent also main geomorphic and geophysical units observed in the field) allow for the simulation of a more flexible movement of the mass. The same material (material 1 in Table 1) was attributed to all landslide blocks. It corresponds
to the average type of the material found within the landslide. The original material of BTL is a gneiss which, by weathering, is partially transformed into a clay on the surface. The depth of the weathered layer is about 20 m. The study area experiences alternations between dry and rainy seasons. The long dry season (from June to September) is followed by the small rainy season (from October to December), then by the small dry season (during the months of January and February). The cycle ends with the strong rainy season from March to May, just before the return of the dry season. Since the photos in Fig. 2 were
taken in October, the ground was relatively wet, but not as wet as it is usually the case in December and during the strong rainy season. Especially for the lower parts of the landslide, the humidity is never very low due to the recharge of the water table by the ponds of water located on the landslide. On the other hand, the groundwater recharge follows the dynamics of the seasons. In the context mentioned above, the action of the rainy season in the body of the landslide is quickly sensible, due to the higher amounts of infiltrating water. Material 2 was attributed to the stable bedrock (Table 1 and Fig. 7).

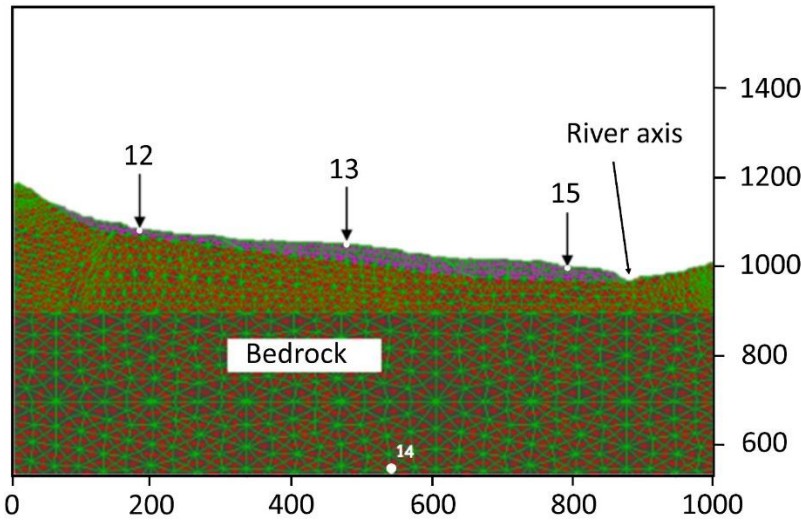


**Figure 7.** Materialization of blocks, joints and materials for the actual model. The history (measurement) points 12, 13 and 15 (white dots) located, respectively, on the upper, middle and lower block correspond to the surface area where parameters were monitored (e.g. the *x*-acceleration). The point 14 is located at the basis of the model, within the bedrock. The axis of the Kanyosha River is located to the right of the history point 15.


The block materials were considered as purely elastic; therefore, the plastic deformation was only computed along joints. For the block materials the following properties were defined: dry density($\rho$), Young's modulus (E), bulk modulus (K) and shear modulus (G), Poisson's ratio ($\nu$), (elastic properties determined on the basis of the estimated and locally measured P-wave velocity, Vp and S-wave velocity, Vs). To allow for plastic deformation along the joints, it is necessary to define the cohesion 200 and the friction angle for the joint/contact material between the blocks. The contact properties are summarized in Table 2. Plastic contact materials were used along the sliding surface and between the blocks (joint material 1); for other (auxiliary) contacts, joint material 2 was used, which only allows for elastic deformation.

**Table 1**. Parameters used for the blocks properties

| Location | Vp (m s$^{-1}$) | Vs (m s$^{-1}$) | $\nu$ | $\rho$ (kg m$^{-3}$) | E (GPa) | K (MPa) | G (MPa) | Material |
|---|---|---|---|---|---|---|---|---|
| Sliding blocks | 1800 | 800 | 0.38 | 2200 | 3.88 | 5250.67 | 1408 | 1 |
| Bedrock | 2600 | 1400 | 0.30 | 2500 | 12.27 | 10366.67 | 4900 | 2 |


**Table 2.** Contact properties: applied values for normal stiffness (jkn), tangential stiffness (jks), range of used cohesion values (jcoh), range of used friction angle values (jfric) and permeability (jperm)

| Contact material | jkn (Pa m$^{-1}$) | jks (Pa m$^{-1}$) | jcoh (MPa) | jfric ($^0$) | jperm |
|---|---|---|---|---|---|
| Joint material 1 | 1000 | 10000 | 0.01-0.05 | 10-20 | 0 |
| Joint material 2 | 1000 | 10000 | 2.00E+20 | 2.00E+20 | 0 |

Scenarios were prepared based on the knowledge of the landslide triggering and evolution factors. Those scenarios were preceded by a back-analysis as the pre-slide topography was used as starting point. Calculations first targeted the reproduction of the present situation of the mass movement before simulating future possible evolutions of the landslide, including the formation of a dam. Variable factors are related to slope geometry, slope material strength, hydrogeological conditions, structural discontinuity, weathering, development of weak zones, lithology and earthquakes (those variables were selected according to those used by published works, such as by Bhasin and Kaynia, 2004, Umrao et al., 2011, Singh et al., 2013a, Kainthola et al., 2012 and Sharma et al., 2017). As major triggering factor, the variable groundwater level was modelled. Further, to test the possible seismic influence on initial slope stability and the possible future evolution of the landslide, a synthetic earthquake signal was used as input for some models.

Actually, a partial contribution of earthquake shaking to the destabilization of the slope is highly probable as the site is located in a seismically active area (see last seismic hazard maps of the Western Branch of the East African Rift by Delvaux et al., 2016). Data availability helps to refer to an existing database and case studies within the study area. Unfortunately, in the context of data scarcity in the region (for instance, there are no strong motion records available for the target area), it is not easy to fix suitable unique values for predictions. This was handled by the use of 4 shaking duration values to well illustrate the behavior of the model corresponding to different scenarios. The seismic context was analyzed on the basis of earthquakes data from the Global Seismographic Network stations of the Incorporated Research Institutions for Seismology (IRIS) on the Lake Tanganyika Region. Therefore, based on that situation, we applied a Ricker wavelet with maximum amplitude of 0.105 $g$ (about 1.05 m s$^{-2}$) and central frequencies of 0.5 and 1.4 Hz. The loading was varied in terms of changing shaking duration. Four different values were considered: 14 seconds, 17 seconds, 25 seconds and 51 seconds. Figure 8 provides the corresponding signals.

The effects of groundwater level were studied considering 5 different cases: no groundwater (dry scenario), saturation of the whole profile (GWT4), groundwater level at a depth of 15 m in the upper block and the saturation of the middle and lower blocks (GWT5), and finally, the groundwater at a regular depth of 7 m below the surface (GWT6). Results discussed in this paper derive from a set of 52 scenarios given in Supplement 2 (Fig. S1).

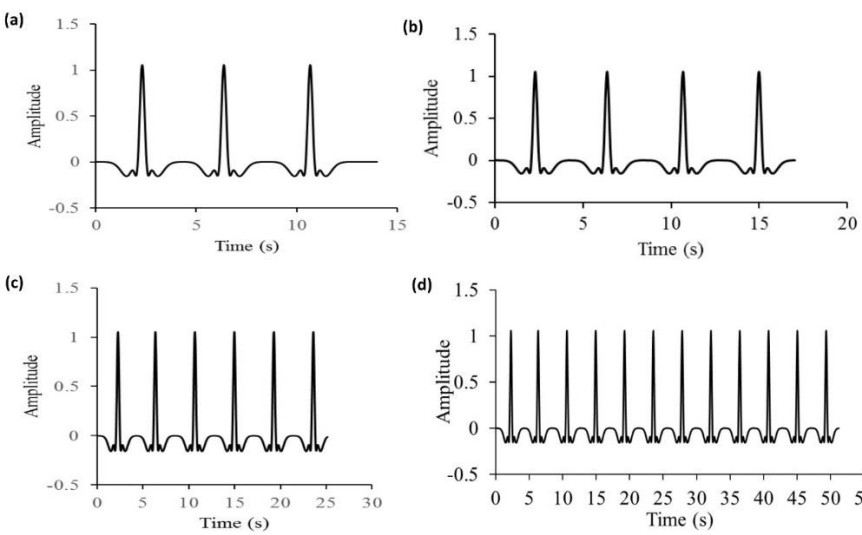

**Figure 8.** Four signals of changing duration and composed of Ricker wavelets (here with normalized amplitude), corresponding to: 14 s (a), 17 s (b), 25 s (c) and 51 s (d), were used as seismic inputs.

## 2.4 Hydraulic modelling

For the hydraulic analysis, we used the academic model WOLF 2D, which solves the shallow-water equations by means of a stable and conservative finite volume scheme. This model has been extensively validated and applied for simulating flow induced by dam and dike breaching (Dewals et al., 2011; Roger et al., 2009) as well as for conducting flood risk analysis (Arrault et al., 2016; Beckers et al., 2013; Bruwier et al., 2015; Detrembleur et al., 2015; Ernst et al., 2010).

We only included water in the flood wave computation, while the actual breaching of the landslide dam would release a substantial amount of solid material. The real flow would have an intermediate behaviour between clear-water flow and debris or granular flow. As shown in Table S2 (Supplement 3), some recent studies neglected sediment transport in the analysis of floods induced by the breaching of landslide dams (Fan et al., 2012; Yang et al., 2013), while others did take sediment transport into account (Li et al., 2011; Shrestha and Nakagawa, 2016) since it may have considerable implications on the volume of mobilized material as well as on morphological evolutions of the valley bottom (e.g., sediment deposition). Nonetheless, we believe that, in the context of the present study, going for more complexity in the modelling framework (i.e. including sediment transport) would mainly produce more speculative results because validation data are neither available for our case study nor for any similar one in the region, which remains largely understudied. Table S2 shows that previous studies which considered sediment transport benefited all from available validation data, such as observed flood discharges or depths of sediment deposits. The implications of this assumption are further discussed in Subsection 4.3.3.

We detail below how friction was parametrized in the hydraulic model, as well as the prescribed boundary conditions and the modelling procedure, including the parametrized breaching mechanism included in the flow simulations.

### 2.4.1 Parametrization of friction

In the hydraulic model, flow resistance was parametrized using the formulation developed by Machiels et al. (2011). Compared to more standard friction formulae (e.g., Manning, Chezy), it offers two main advantages: *(i)* being truly physically-based, it reduces substantially the need for recalibrating the model when the range of flow rate is varied; *(ii)* the only parameter to be set is the characteristic size of bottom irregularities, which can be estimated from field observations. This parametrization is hence particularly suitable for applications for which only scarce flow monitoring data are available, such as in the present case.

Here, we tested three values for the roughness height: 0.1 m, 0.2 m and 0.3m, corresponding to the prevalent class of grain size in the riverbed material (as described in section 2.1). In the following, to show the effects of the roughness of the river bed, we present the results for the two extreme values of the roughness height ($k_s = 0.1$ m and $k_s = 0.3$ m).

### 2.4.2 Hydraulic boundary conditions and computed scenarios

The upstream boundary condition is a prescribed flow hydrograph, representing either a flood wave coming from the upstream catchment or a steady inflow. As detailed in Subsection 2.4.3 and in Supplement 4, we used only steady inflows, corresponding respectively to a base flow (3 m$^3$ s$^{-1}$), a 20-year flood (peak discharge of 60 m$^3$ s$^{-1}$) and a 50-year flood (peak discharge of 120 m$^3$ s$^{-1}$).

At the downstream end of the computational domain, the river mouth in Lake Tanganyika was not included explicitly because only limited information was available on bathymetric and hydraulic data at this location. Consequently, the hydraulic behaviour of the river mouth is lumped into the boundary condition prescribed at the downstream end of the computational domain.

The proposed boundary condition is based on a weir equation, relating the outflow discharge $Q_o$ to the averaged water level $h_o$ close to the simulation downstream boundary:

$$Q_0 = \frac{2}{3} C_D L \sqrt{2g(h_0 - w)} \qquad (1)$$

with $g$ being the gravity acceleration (m s$^{-2}$), $C_D$ a non-dimensional discharge coefficient (taken equal to 0.75), $L$ an equivalent crest length (m) and $w$ an equivalent crest height (m). Equation (1) enables simulating different configurations (e.g., loosely vs. strongly varying downstream water level when the flow rate changes) and we performed a sensitivity analysis by varying $L$ and $w$. For very high values of $L$, $h_o$ remains virtually constant whatever $Q_o$; otherwise it varies with $Q_o$. However, as shown in Supplement 5, this boundary condition has actually an influence only over a very limited distance upstream of the domain boundary: in all the conducted tests, this influence zone did not extend over more than 300 meters. This very limited influence results from the relatively steep slope of the river (around 1.5 % in the downstream area; 6 % in the upstream reach). Consequently, the particular formulation of the downstream boundary condition (Eq. (1)) can be safely disregarded when analysing the modelling results over virtually the whole computational domain (except the most downstream 300 m) since they remain independent $L$ and $w$.

### 2.4.3 Modelling procedure

The hydraulic simulations aim at evaluating the impact of the dam failure as a result of the water impoundment behind it and the river overflowing the dam crest. Thus, the initial step of hydraulic modeling considers a filled reservoir and a steady flow of water over the crest of the dam before failure. In line with Dewals et al. (2011), the modelling procedure involves two steps (Table 3):

- step 1: a pre-failure steady flow is computed in the river, under three different hydrological scenarios (steady flow corresponding to the mean discharge in the river or to a 20-year flood, or a 50-year flood);
- step 2: using the result of step 1 as initial condition, the flow induced by the breaching of the dam is computed.

In Step 1, the dam geometry is incorporated in the topographic data used for flow computation. This means that the dynamics of material sliding into the river is not explicitly reproduced in the hydraulic modelling. As it is not possible to anticipate when the landslide dam breaching might occur, we consider three different pre-failure flow conditions: base flow, 20-year flood and 50-year flood. In Step 2, using a parametric description of the breaching, the dam is gradually removed from the topography, so that the water impounded behind the dam is released. The model computes the unsteady propagation of the induced flood wave in the downstream valley.

Examples of results of Step 1 and Step 2 are displayed respectively in Fig. S2 and Figs S3 to S6 in Supplement 6.

**Table 3.** Two-step hydraulic modelling protocol

|        | Hydraulic computation    | Dam                                                     |
|--------|--------------------------|---------------------------------------------------------|
| Step 1 | Steady-state simulation  | Incorporated in the DEM used for the simulation         |
| Step 2 | Unsteady simulation      | Gradually removed from the DEM (time-dependent topography) |

### 2.4.5 Modelling of the breaching mechanism

The mechanisms of breaching of natural dams are complex, highly variable and incompletely understood. Hence, the modelling
of the dam breaching may be a substantial source of uncertainty. In the present study, process-oriented modelling of the
breaching was not considered as a viable option, mainly due to the lack of detailed information on the dam material (graded,
non-homogeneous material), the complexity of the breaching of natural dams and the absence of validation data from similar
case studies in the region. Instead, we opted for a simpler *parametric description* of the dam breaching which appears more
consistent with the quality of available data and the overall level of uncertainty affecting the present study.

Among the various possible failure modes, we chose to represent dam *overtopping*, which is the most frequent failure mode
for landslide dams. Failure induced by dam overtopping was reported for over 90 % of all landslide dams reviewed by Costa
and Schuster (1988) and for 131 out of 144 cases reviewed by Peng and Zhang (2012).

As sketched in Fig. 9, the parametric breach model was implemented in the 2D flow model by means of a time varying
topography. The breach outflow is thus explicitly computed by the flow model, enabling the representation of the hydraulic
coupling between reservoir depletion, flow through the breach and possible backwater effects. This procedure requires a user-
defined initial dam geometry (Fig. 9a) and a user-defined final geometry corresponding to the breached dam (Fig. 9e). In-
between these two geometries, the algorithm performs a linear interpolation in time (Dewals et al. 2011). The breaching
duration also needs to be prescribed by the user.

Several prediction formulae have been tested for estimating the breaching duration (Froehlich 2008, Peng and Zhang 2012,
BREACH model …). They lead to scattered values, ranging in-between 10 min and one or two hours. Such discrepancies
result from the limited number of real-world case studies for which information on breaching duration is available. For
instance, out of a total of 1,239 cases reported by Peng and Zhang (2012), only 52 contain detailed information on the breaching
and only 14 cases have records of breaching duration. Moreover, inconsistencies exist in these records, so that the regression
results for breaching duration are generally less satisfactory (in terms of $R^2$) than for other breach parameters. These are the
325 reasons why we considered a range of plausible assumptions on the breaching duration, in-between 10 min and 1 h. We also
tested one extreme assumption (instantaneous dam failure) to characterize the envelope of possible results. The latter scenario
could also correspond to an almost instantaneous breaching as a result of an earthquake.

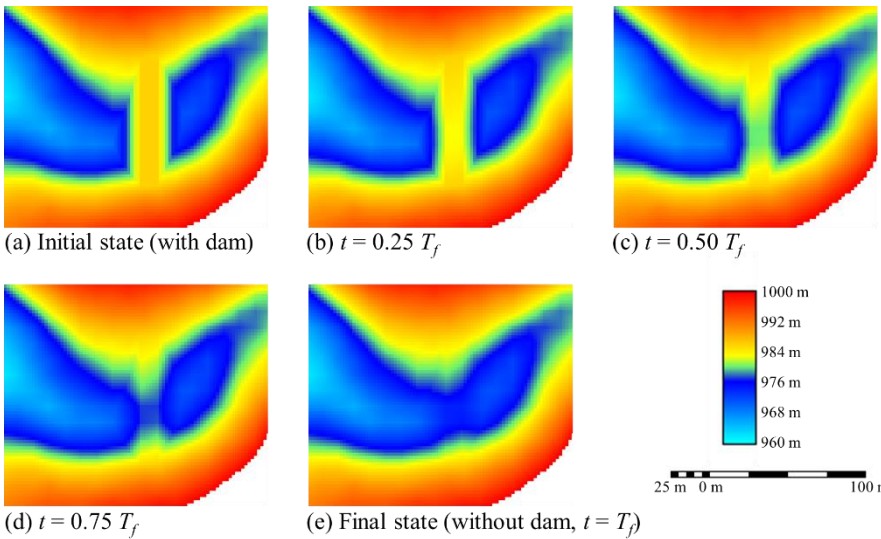

(a) Initial state (with dam)    (b) $t = 0.25\ T_f$    (c) $t = 0.50\ T_f$

(d) $t = 0.75\ T_f$    (e) Final state (without dam, $t = T_f$)

**Figure 9.** Plane view of the topography evolution in the near-field of the landslide dam as a function of time ($T_f$ stands for the breach formation time).

## 2.5 Flood intensity mapping

The results of the hydraulic computations were processed to display the inundation extent as well as information on water depth and flow velocity in the floodplains. The method used by Alvarez et al. (2017) was considered for the classification of flood intensity in high, medium and low categories. To be classified in the high category, the location must have a water depth higher than 1 m, a water velocity greater than 1 m s$^{-1}$ or a product of the velocity and the water depth greater than 0.5 m$^2$ s$^{-1}$. Conditions to be classified in the category of low flood intensity are: a water height below 0.5 m, a flood velocity below 0.5 m s$^{-1}$ and a product of the velocity and the water depth below 0.25 m$^2$ s$^{-1}$. The medium intensity category corresponds to all intermediate situations.

## 3 Results

### 3.1 Landslide triggering: back analysis

The results obtained from the elastic model with initial topography (scenarios 1 and 2 in Supplement 2, Fig. S1a) were first measured in terms of peak ground acceleration (PGA) and Arias Intensity (Ia, see Arias, 1970) in different parts of the profile. This was calculated from the acceleration recorded in x-direction for specific history points chosen within the model profile. Figure 10 and Table 4 provide x-acceleration, PGA and Ia for the upper and for the lower blocs considering 14 seconds and 25 seconds. As we were interested in finding how the landslide was triggered and evolved, we tracked the upper block displacement and its detachment from the later scarp, while the lower block movements needed to be analysed in detail to assess the damming potential (also in comparison with the present situation).

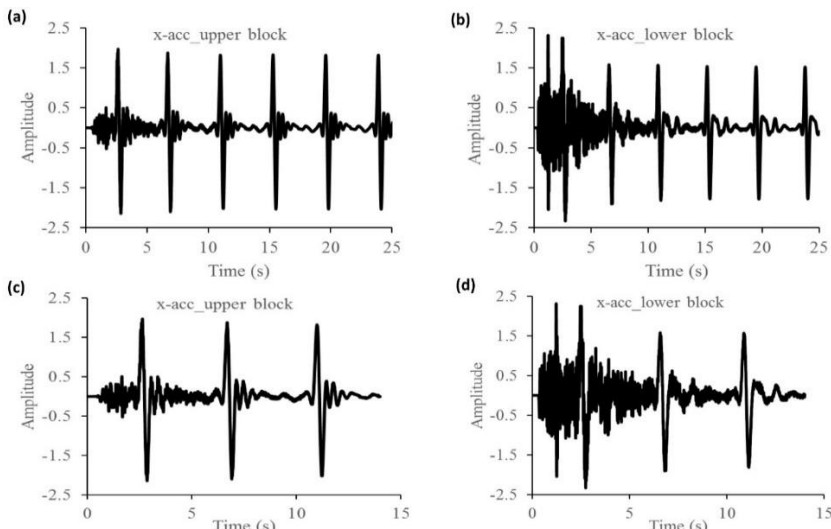

**Figure 10.** X-acceleration for 25 (a, b) and 14 (c, d) seconds computational time. Accelerations labelled as 'xacc_lower block' and 'xacc_upper block' correspond respectively to history points 15 and 12 mentioned in Fig. 7.

**Table 4.** PGA and Ia measured along the profile for the 14 and 25 seconds in the bedrock, in the upper block (point 12*), in the middle block (point 13*) and in the lower block (point 15*).

| Location | Duration=14sec | | Duration=25sec | |
|----------|----------------|----|----------------|----|
| | PGA, $(m\ s^{-2})$ | Ia $(m\ s^{-1})$ | PGA $(m\ s^{-2})$ | Ia $(m\ s^{-1})$ |
| Rock | 1.99 | 0.55 | 1.99 | 0.28 |
| 12* | 1.97 | 0.95 | 1.97 | 0.48 |
| 13* | 2.24 | 1.05 | 2.24 | 0.55 |
| 15* | 2.42 | 0.91 | 2.31 | 0.52 |

Regarding the main landslide triggering factors, this was assessed by analysing the calculated safety factor. Scenarios were simulated to highlight the intrinsic behaviour of the model under different loading conditions. First this was fulfilled in the absence of water and seismic loads. Then, groundwater was added to the model and a seismic input was used. The groundwater data were recorded along the sliding surface with an x-increment of 10 meters. Results of the safety analysis was completed for different hydrogeological conditions.

Dry and non-seismic models are assumed to be much more stable. Therefore, scenarios have been made to track the limits from which instability begins. Our discussion is based on the results of the safety factor obtained for a cohesion of 0.01 MPa and 0.02 MPa and for friction angles of 15 °, 17 ° and 20 ° as summarized in Table 5.

As expected, those results in Table 5 show a strong dependence of the Factor of Safety (FoS) of the slope on the friction angle of the slope material. Furthermore, we notice that the FoS of the slope for dry and non-seismic scenarios is almost twice larger than the safety factor corresponding to saturated and seismic conditions. Actually, in the absence of water and seismic
vibration, the initial slope of the Banana Tree Landslide site would have been stable unless very (and unrealistically) low values of cohesion and friction angle are considered (e.g. friction angle of less than 10°). This confirms our first estimates of the important role of groundwater pressures and seismic vibrations with respect to the slope destabilization. Based on the local and regional context, other environmental and anthropogenic parameters were identified as factors that have contributed to the increase of field stresses, forcing the landslide triggering and evolution. These factors are: earthquakes, erosion at the slope
toe (fluvial erosion and quarrying) and upper slope overloading due to the installation of the inhabitants. The last also causes other effects like the vegetation removal and galleries due to some cultural technics which can evolve to a favourable situation to landslide triggering under heavy rain context. This is in line with steps of the process leading to slope instability and landslide triggering as described by Terzaghi (1950), Varnes (1978), Popescu (1994) and Popescu (2002). Moreover, the general north-south direction of the layers could have contributed much to the process amplification. As illustrated in Fig. 2b, the layers are
parallel to the direction of the sliding, this allows easy movements downwards in case of even small slope destabilization.

**Table 5.** Safety factor obtained for a cohesion of 0.01 MPa and 0.02 MPa for different friction angles (G1=dry and non-seismic; G4 is seismic and saturated). Scenarios involving groundwater and seismic shaking considered a complete saturation of the sliding layers (additional GWT5 scenario) and a wavelet of 25s shaking time. G4a and G4b correspond to the partial and complete saturation.

| Joint cohesion (MPa) | Joint Fric. angle (°) | Safety factor/G1 (-) | Safety factor/G4a (-) | Safety factor/G4b (-) | Ratio $\dfrac{G1}{G4a}$ | Ratio $\dfrac{G1}{G4b}$ |
|---------|-----|------|------|------|------|------|
| | 15 | 1.59 | 0.89 | 0.81 | 1.79 | 1.96 |
| 0.01 | 17 | 1.68 | 1.03 | 0.91 | 1.63 | 1.85 |
| | 20 | 2.23 | 1.22 | 1.09 | 1.83 | 2.05 |
| 0.02 | 17 | 1.75 | 1.05 | 0.95 | 1.67 | 1.84 |

### 3.2 Analysis of the actual state of stability and potential $x$-displacement

After the back-analysis, simulations of the current situation of the landslide were computed to study the present landslide state of stability. In this section, we have focused on a displacement-oriented analysis, as the main purpose is the study of the conditions under which the landslide could form a dam.The results given in Fig. 11 and Table 6 constitute the basis for this analysis. Large PGA and Ia are observed at the lower (and thicker) block of the landslide. This difference is also observed for the values and the distribution of the x-accelerations during the shaking time, again with high values for the downstream block.

This difference will also affect the disproportionate horizontal x-displacements of the blocks, creating extension and compression zones. Extension zones can lead to the opening of large cracks.

**Table 6.** PGA and Ia in the profile for the 14 and 25 seconds. Locations 12\*, 13\* and 15\* refer to the upper, the middle and the lower bloc, as mentioned in Fig. 7

| Location | Duration=14sec | | Duration=25sec | |
|---|---|---|---|---|
| | PGA, $(m\ s^{-2})$ | Ia $(m\ s^{-1})$ | PGA $(m\ s^{-2})$ | Ia $(m\ s^{-1})$ |
| Rock | 1.97 | 0.30 | 1.90 | 0.57 |
| 12* | 3.34 | 1.08 | 2.04 | 0.71 |
| 13* | 2.12 | 0.52 | 2.10 | 1.32 |
| 15* | 4.24 | 1.17 | 2.45 | 1.10 |


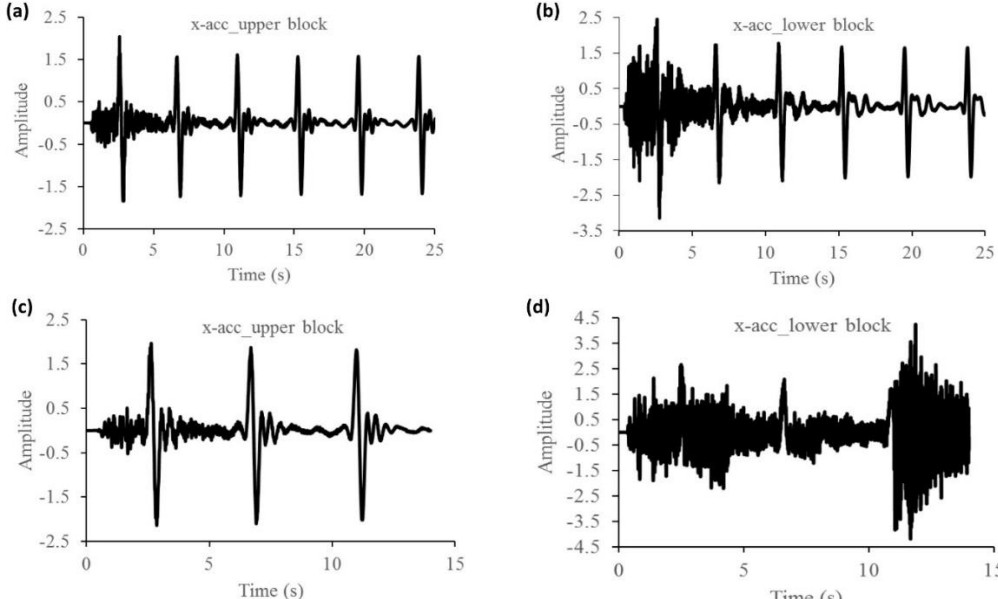

**Figure 11.** Actual- X-acceleration for 25 seconds (a, b) and 14 seconds (c, d) computational time.

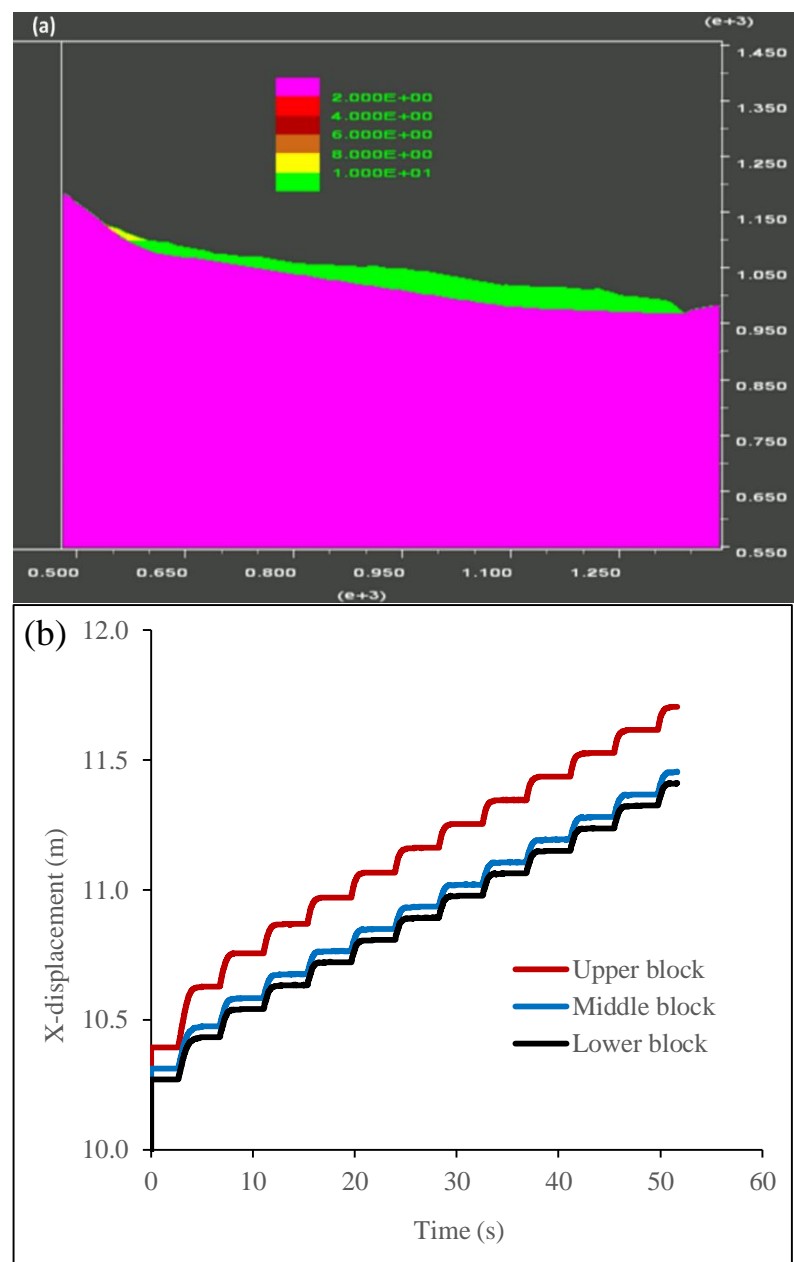

**Figure 12.** Plots of blocks (a) and displacements (b) as given in the UDEC output for run 13 (see Supplement 2, Fig. S1b: using a cohesion of 0.01 MPa and a seismic shaking of 51 seconds).

Figure 12 describes the landslide situation after scenario 13 (detailed in Fig. S1b), showing that increasing the shaking duration would result in a displacement increase over 12 m. The model sometimes provided disproportionate displacements between the three main blocks (Fig. 12b). This leads to compression and shear zones between the blocks and could even probably be the main cause of the spurts of ground water to form the small lakes hanging up the Kanyosha River.

**Table 7.** The role of water in the model behaviour. Values are based on recorded displacements under scenario 9, 10, 11, 14, 16 and 17.

| | Friction angle=11°<br>t=25 S, c=0.01<br>$\dfrac{saturated}{Dry}$ | Friction angle=15°<br>t=25 S, c=0.01<br>$\dfrac{saturated}{Dry}$ | Friction angle=17°<br>t=25 S, c=0.01<br>$\dfrac{saturated}{Dry}$ |
|---|---|---|---|
| Xdis_UPPER Bloc | 14.2 | 7.8 | 3.0 |
| Xdis_LOWER Bloc | 5.1 | 2.4 | 2.9 |

The results of this Table 7 show the effects of water on the dynamics of the BTL. Under certain conditions of cohesion and shaking duration, the presence of water increases the X-displacement by 2.4 to 14 times. Kainthola et al. 2012 found a change of 79.1 %, corresponding to an increase of approximately 1.8 times. This explains why many cases of reactivation or acceleration of landslides occur during rainy periods. These results are discussed with more detail in section 4.1. A full river blockage is possible. Actually, it is likely that the displacements would have been larger for stronger shaking and if we had also modelled plasticity within the blocks. Furthermore, we must consider that some destabilisation mechanisms cannot be computed with UDEC, such as fluidisation or liquefaction of the clayey landslide material, which would produce much larger displacements.

## 3.3 Effects of the dam breaching on flood intensity

### 3.3.1 Water depth

In this section, we examine to which extent the water depths are affected by the occurrence of a landslide dam breaching. The computed water depths are discussed here for four cross-sections, labelled sections 1 to 4 (Fig. 3). Figure 13 displays the computed water depths for the pre-failure flow and for the breach-induced flow, for section 1 to 4, considering a roughness height of 0.1 m (Fig. 13a) and 0.3 m (Fig. 13b), as well as three pre-failure flow scenarios (base flow, 20-year flood and 50-year flood).

The results strongly depend on the assumed breaching time, pre-failure flow scenario and distance to the dam, whereas the values of $k_s$ has a more limited influence on the results.

In the extreme case of an instantaneous failure, the computed water depth in section 1 is about 24 times higher when instantaneous dam breaching is assumed compared to a base flow situation without dam breaching. This value is reduced to about 5 and 4 respectively for pre-failure flow conditions corresponding to a 20- and a 50-year flood. Similarly, the increase in water depths induced by the instantaneous dam breaching becomes more moderate for sections 2, 3 and 4 which are located respectively at about 2, 4 and 6 km downstream of the dam. In the case of a 20-year flood or a 50-year flood, the maximum water depth is less than doubled in section 3 (+ 50 − 70 %) and 4 (+ 20 − 30 %).

In the case of a gradual dam failure in 10 minutes, dramatic increases in water depths are obtained only in the case of a base flow as pre-failure flow scenario. In such a case, the computed water depths are multiplied by approximately 9, 5 4 and 3 in sections 1, 2, 3 and 4, respectively. In contrast, in the cases of a 20- or 50-year flood as initial flow conditions, the computed water depths are, at maximum, about doubled. In section 4, the increases are limited to 20 − 30 %. Hence, the severity of the amplification of water depths as a result of dam breaching is, in relative terms, significantly influenced by the assumed pre-failure flow.

Finally, in the case of a gradual breaching in 60 minutes, the computed water depths are affected by a factor of 3.6 in section 1 and 2.1 − 2.6 in sections 2 to 4 if a base flow is assumed as initial condition. In contrast, if a 20- or a 50-year flood is assumed initially, the growth in the computed water depths as a result of dam breaching is generally no more than about 20 %.

Nonetheless, in all cases, the increases in water depth as a result of dam breaching remains highly significant from the perspective of flood risk. These results show that dam breaching exacerbates considerably the flood conditions in the downstream river. This conclusion remains robust despite the high uncertainties on the roughness parameter. Indeed, as shown in Table 7, changing the roughness has little influence on the relative effect of dam breaching on the water depths. This is also confirmed by the high similarity between Figs. 13a and 13b.

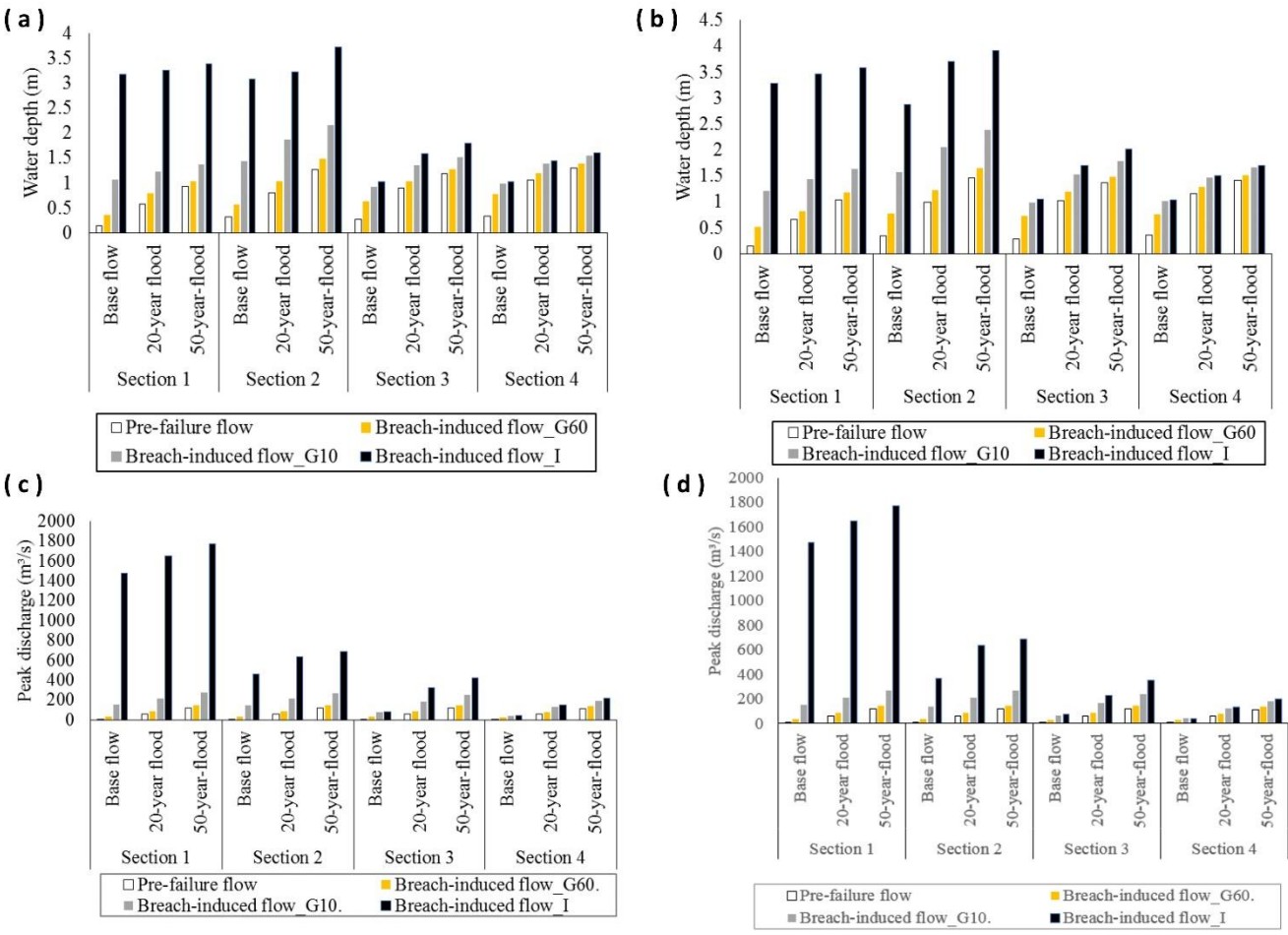

**Figure 13.** Computed maximum water depths (a, b) and peak discharges (c, d) in cross-sections 1 to 4, for various pre-failure flow conditions (base flow, 20- and 50-year floods) and for a roughness height $k_s = 0.1$ m (a, c) or 0.3 m (b, d). 'Breach-induced flow_G60', 'Breach-induced flow_G10' and 'Breach-induced flow_I' stand for 'flow induced by the failure of the landslide dam with, respectively, a breaching time of 60 minute, a breaching time of 10 minute and an instantaneous failure.

**Table 8.** Ratio between the maximum water depth ($H_{max}$) following dam breaching and the water depth in the pre-failure flow conditions in sections 1 to 4, considering two different roughness heights ($k_s = 0.1$ m and $k_s = 0.3$ m) and various pre-failure flows (base flow, 20-year flood and 50-year flood). I and $G_{10}$ and $G_{60}$ stand for instantaneous, 10 minutes-gradual breaching and 60 minutes-gradual breaching respectively.

| Section | ks | Failure mode | $H_{max}$ ratio | | |
|---|---|---|---|---|---|
| | | | Base flow | 20-year flood | 50-year flood |
| Section 1 | 0.1 | I | 22.60 | 5.70 | 3.60 |
| | | $G_{10}$ | 7.57 | 2.16 | 1.47 |
| | | $G_{60}$ | 2.57 | 1.39 | 1.12 |
| | 0.3 | I | 23.50 | 5.30 | 3.50 |
| | | $G_{10}$ | 8.64 | 2.18 | 1.58 |
| | | $G_{60}$ | 3.64 | 1.24 | 1.15 |
| Section 2 | 0.1 | I | 9.60 | 4.00 | 2.90 |
| | | $G_{10}$ | 4.50 | 2.34 | 1.69 |
| | | $G_{60}$ | 1.75 | 1.29 | 1.17 |
| | 0.3 | I | 8.50 | 3.70 | 2.70 |
| | | $G_{10}$ | 4.62 | 2.07 | 1.64 |
| | | $G_{60}$ | 2.26 | 1.23 | 1.12 |
| Section 3 | 0.1 | I | 3.80 | 1.80 | 1.50 |
| | | $G_{10}$ | 3.41 | 1.53 | 1.29 |
| | | $G_{60}$ | 2.37 | 1.17 | 1.08 |
| | 0.3 | I | 3.80 | 1.70 | 1.50 |
| | | $G_{10}$ | 3.54 | 1.50 | 1.31 |
| | | $G_{60}$ | 2.57 | 1.18 | 1.08 |
| Section 4 | 0.1 | I | 3.10 | 1.40 | 1.20 |
| | | $G_{10}$ | 2.97 | 1.30 | 1.18 |
| | | $G_{60}$ | 2.33 | 1.12 | 1.06 |
| | 0.3 | I | 3.00 | 1.30 | 1.20 |
| | | $G_{10}$ | 2.91 | 1.26 | 1.17 |
| | | $G_{60}$ | 2.14 | 1.11 | 1.06 |

### 3.3.2 Peak discharge

The peak discharge of the flood waves induced by instantaneous dam breaching are in the ranges 1500 -1700 m³ s⁻¹, 460-570 m³ s⁻¹, 77-300 m³ s⁻¹ and 41-110 m³ s⁻¹ in sections 1, 2, 3 and 4 respectively (Figs. 13c and 13d). In the uppermost section (n°1), which is located close to the toe of the dam, the roughness height has virtually no influence on the computed peak discharge as the flow in this area is predominantly controlled by the dam failure. In contrast, the peak discharge is gradually more influenced by the roughness height as the flood wave propagates towards the more downstream cross-sections 2, 3 and 4. Similarly to the results for the water depths, the peak discharges decrease significantly in case of gradual failure; e.g. for a one-hour breaching scenario, these peak flow value ranges become 33.4.-149 m³ s⁻¹, 33.3-148.4 m³ s⁻¹, 28.5-147.2 m³ s⁻¹ and 26.6-134.6 m³ s⁻¹ in sections 1, 2, 3 and 4 respectively. Intermediate results are obtained for a breaching time of 10 minutes.

In cross-section 1, the peak discharge of the instantaneous dam-breaching flood wave is roughly 500 times higher than the base flow, 30 times larger than a 20-year flood and 15 times larger than a 50-year flood (Table 9). In the more downstream cross-sections, these numbers become smaller; but the peak flow after dam breaching remains at least two to ten times larger than typical flood discharges (20- or 50-year floods) and can be 100 times larger than the base flow in the river. These results are only slightly affected by a change in the roughness height.

We find again that neglecting dam failure would result in a strong underestimation of the downstream flood intensity. This underestimation is particularly severe in the cross-sections located close to the dam, whereas in the more downstream area, this effect is mediated by peak flow attenuation during wave propagation.

**Table 9.** Ratio between the peak discharge following dam breaching and the discharge in the pre-failure flow conditions in sections 1 to 4, considering two different roughness heights ($k_s = 0.1$ m and $k_s = 0.3$ m) and various pre-failure flows (base flow, 20-year flood and 50-year flood).

| Section | $k_s$ | Failure mode | $Q_{max}$ ratio | | |
|---|---|---|---|---|---|
| | | | Base flow | 20-year flood | 50-year flood |
| Section 1 | 0.1 | I | 490.0 | 28.0 | 15.0 |
| | | $G_{10}$ | 51.5 | 3.5 | 2.3 |
| | | $G_{60}$ | 11.1 | 1.5 | 1.5 |
| | 0.3 | I | 490.0 | 28.0 | 15.0 |
| | | $G_{10}$ | 51.6 | 3.5 | 2.3 |
| | | $G_{60}$ | 11.1 | 1.5 | 1.2 |
| Section 2 | 0.1 | I | 150.0 | 11.0 | 5.7 |
| | | $G_{10}$ | 47.6 | 3.5 | 2.2 |
| | | $G_{60}$ | 11.1 | 1.5 | 1.2 |
| | 0.3 | I | 120.0 | 11.0 | 5.7 |
| | | $G_{10}$ | 45.3 | 3.5 | 2.2 |
| | | $G_{60}$ | 10.9 | 1.5 | 1.2 |
| Section 3 | 0.1 | I | 27.0 | 5.4 | 3.5 |
| | | $G_{10}$ | 24.7 | 3.0 | 2.1 |
| | | $G_{60}$ | 9.5 | 1.5 | 1.2 |
| | 0.3 | I | 25.0 | 3.8 | 2.9 |
| | | $G_{10}$ | 20.9 | 2.7 | 2.0 |
| | | $G_{60}$ | 9.0 | 1.5 | 1.2 |
| Section 4 | 0.1 | I | 15.0 | 2.6 | 2.0 |
| | | $G_{10}$ | 15.5 | 2.3 | 1.7 |
| | | $G_{60}$ | 8.9 | 1.4 | 1.2 |
| | 0.3 | I | 14.0 | 2.2 | 1.8 |
| | | $G_{10}$ | 13.0 | 2.0 | 1.6 |
| | | $G_{60}$ | 8.1 | 1.4 | 1.2 |

### 3.3.3 Wave propagation time

Figure 14 displays the wave propagation time in sections 1 to 4, i.e. the time elapsed between the dam failure and the moment the flood wave reaches the corresponding section of the river. The time-to-peak, i.e. the time between the dam breaching and the arrival of the peak discharge in the corresponding river sections is also displayed. Results are shown for two roughness heights, $k_s = 0.1$ m and $k_s = 0.3$ m.

In the upper part of the river, the wave propagation time remains mostly independent of the pre-failure flow. The flood wave takes between 2.5 and 3 min to reach section 2, which corresponds to a wave velocity of the order of 10 to 12 m s$^{-1}$. Further 510 downstream (urbanized area), the pre-failure flow has a strong influence on the wave propagation velocity. When the pre-failure conditions in the river correspond to base flow, the wave takes roughly 12 min to reach section 3 and 25 min to reach section 4. These values drop to 7-8 min and 12-14 min if the instantaneous dam breaching takes place during a river flood, corresponding to a rise in the mean wave velocity from 4-6 m s$^{-1}$ in base flow conditions up to 7-9 m s$^{-1}$. In case of a 10-min

gradual breaching, the wave propagation time to get to sections 3 and 4 becomes 9-10 min and 14-16 min respectively. From a 10-min to a 60 min breaching scenarios, the wave travel time is moderately increased by 26% in section 3 and 33% in section 4 when a river flood is considered, but here again, these values remain lower than the corresponding travel time in case of base flow scenarios.

Hence, the higher the pre-failure discharge in the river, the shorter the wave propagation time and time-to-peak. Compared to a dam failure occurring when the river discharge is low (base flow), the wave propagation time and time-to-peak are approximately reduced by a factor two if the failure occurs during a flood, which corresponds incidentally to the most likely scenario. Although dam breaching has a relatively weaker influence on maximum water depth and peak discharge when the pre-failure flow corresponds to flood conditions (sections 3.3.1 and 3.3.2), the results obtained here demonstrate that even in flood situations, dam breaching is particularly dangerous because of the shorter time between the occurrence of failure and the wave arrival. Overall, the velocity of the flood wave gives little chance for the population to take precautionary measures such as evacuation; unless the population is very well prepared and some early-warning system can be put in place.

Figure 14 shows also the diffusion of the flood wave as it propagates in the valley. While the difference between the wave arrival time and the time-to-peak is low in sections 1 and 2 (generally below 0.5 min), it reaches 1 to 2 min in section 3 and 2.5 to 4.5 min in section 4. This shows that the flood wave is considerably steeper in the upper part of the valley (sections 1 and 2). Also, the wave remains steeper when dam breaching occurs during a river flood than when it occurs during base flow.

The value chosen for the roughness height has virtually no influence on the computational results in sections 1 and 2, which are relatively close to the dam; whereas it has more influence at sections 3 and 4. Nonetheless, the main observations detailed above remain valid for both values of the roughness height ($k_s = 0.1$ m and $k_s = 0.3$ m).

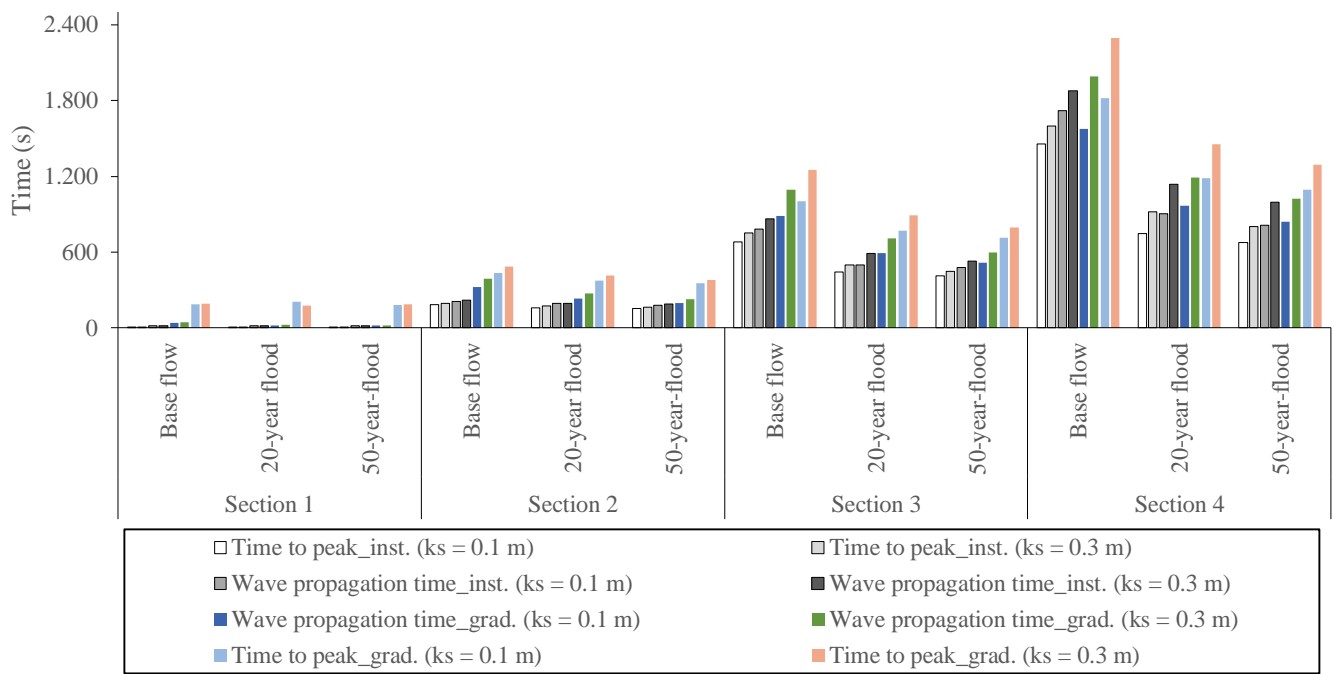

**Figure 14a.** Computed wave propagation time and time-to-peak in sections 1 to 4, for various pre-failure flow conditions (base flow, 20- and 50-year flood) and for two different roughness heights ($k_s = 0.1$ m and $k_s = 0.3$ m). The gradual failure time is 10 minutes.

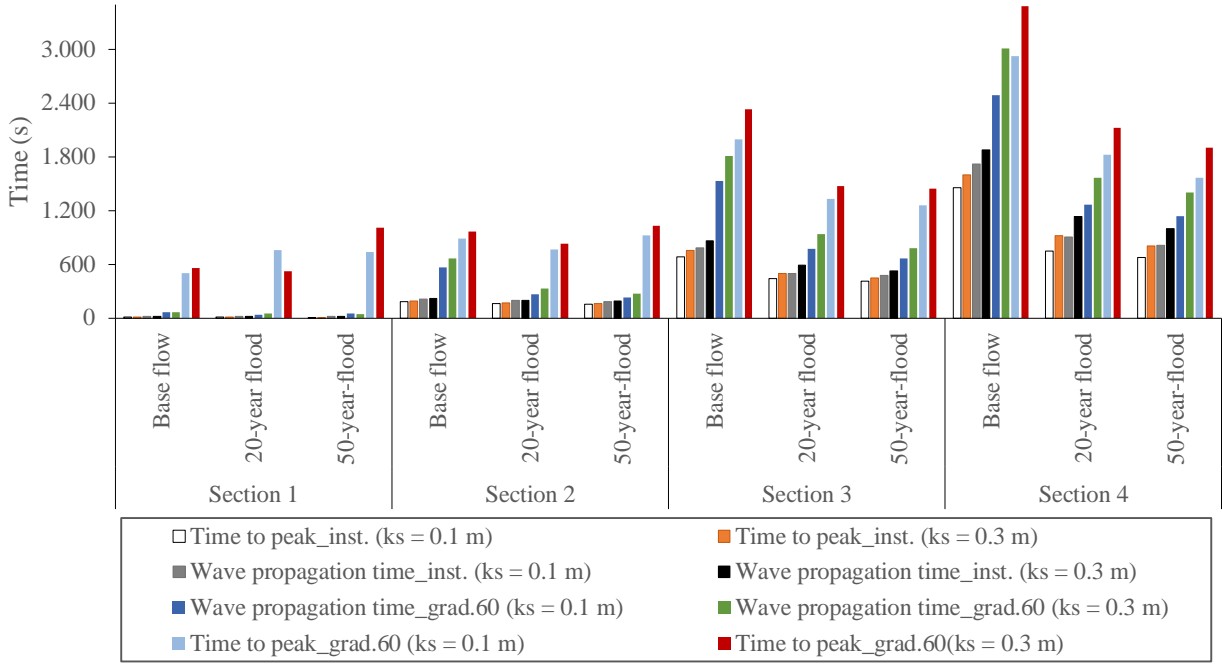

**Figure 14b.** Computed wave propagation time and time-to-peak in sections 1 to 4, for various pre-failure flow conditions (base flow, 20- and 50-year flood) and for two different roughness heights ($k_s$ = 0.1 m and $k_s$ = 0.3 m). The gradual failure time is 60 minutes.

### 3.4 Floodplain delineation and flood intensity mapping

The spatial extent of floodplains, expressed in terms of surface and its variation for different return periods, is analysed here. For each case, as given in Table 10, values are given for both failure and non-failure scenarios. Changes induced by the instantaneous, as well as the 10-min and 60-min gradual dam failure are also quantified and discussed. Under the same roughness height, both in a failure or a non-failure situation, the flood extent remains greatly linked to the steady flow discharge. For example, from the base flow to the 50-year flow, the average flood area increase is 25 %, using a roughness height of 0.1 m. This increase is approximately 16 % from the base flow to the 20-year steady flow. These ratios remain almost constant both in the failure and non-failure scenarios.

The floodplain extent variations are also linked to the roughness changes. For pre-failure scenarios, from a height of 0.1 m to 0.3 m, the surface of the floodplain increases by 10 %, 14 % and 34 % for the base flow, the 20-year and 50 years of return period. These increases are 4%, 8% and 29%; 4%, 9% and 21%; 6 %, 7 % and 19 % in case of a 60-min dam breaching, a 10-min dam breaching and an instantaneous dam breaching.

The maps in Fig. 15a and Fig. 15b show the spatial distribution of the flood intensity. Values are calculated on the basis of the water depth and velocity. Then, they are classified according to the methodology described in Section 2.5. The maps show the impact of the dam failure on the flood intensity. These maps relate to the lower parts of the watershed, between sections 3 and 4 of Fig. 3, in the city of Bujumbura. For both Fig. 15a and Fig. 15b, maps in the first column (left column) represent the scenarios without dam breaching while those in the third column relate to the corresponding instantaneous failure scenarios.

The maps in the second column correspond to the intermediate situation: gradual failure in 10 min for Fig. 15a and 60 min for Fig. 15b. Subfigures (a), (b) and (c) relates to pre-failure flow conditions corresponding to base flow, while the subfigures (d) to (f) are related to a pre-failure 50-year flood with a roughness height of 0.1 m. Subfigures (g), (h) and (i) differ from those in the second rows by the fact that a roughness of 0.3 m is applied instead of 0.1 m. The comparison between maps of the first and second rows helps to analyse changes related to the initial flow, while the differences between the second and the third rows are the result of the change in roughness within the bottom of the river. Each time, the maps in the second and third columns highlight changes due to the dam breaching.

The maps in the first row correspond to the base flow case. Their comparison allows to realize a significant change especially downstream with a lateral extension of the flooded area. Thus, notable changes are observed and consist of a change in the flood intensity level. According to subfigures (a), (b) and (d), almost all zones classified in the low level flood intensity category in the non-failure case migrated directly into the high flood intensity category in case of a failure scenario [(a) and (b)]. This is also the case from the base flow to a 50-year flow [(a) and (d)] but here, the change due to the increase of pre-failure flow is more important than that resulting from the dam breaching. The vertical comparison between the first two rows highlights the variations of the flood intensity depending on the initial flow rate, as well as in a failure and in a non-failure case, under a roughness height of 0.1 m. Unlike the previous ones [(a) and (b)], the no-breach scenario (Fig. 15d) already includes zones under the high-category flood intensity. However, the lateral extension of flooding is much more obvious than previously, especially near cross-section 3. The corresponding failure scenario [map (e)] shows significant increases in flood intensity both on the south and north river banks. Comparison between (d), (e) and (f) to maps (g), (h) and (i) reveals that a higher roughness height increases substantially the estimated flood intensity, due to the corresponding increase in water depth. These observations apply to both Figs. 15a and 15b. The main difference observed between Figs. 15a and 15b relates to subfigures (b), (e) and (h) corresponding to the gradual failure. The flood intensity is higher for a breaching time of 10 min (Fig. 15a) than for a breaching time of 60 min (Fig. 15b). Overall, the flood intensity increases as the pre-failure flow increases and as breaching time becomes shorter.

**Table 10.** Predicted change in terms of flooded area due to the landslide induced dam breaching for roughness = 0.1 m and 0.3 m.

| Pre-failure flow | Pre-failure flooded area (m²) | Flooded area after dam failure (m²) | | | Relative increase in flooded area as a result of dam breaching (%) | | |
|---|---|---|---|---|---|---|---|
| | | Instantaneous | Gradual (10 min) | Gradual (60 min) | Instantaneous | Gradual (10 min) | Gradual (60 min) |
| Roughness height $k_s = 0.1$ m | | | | | | | |
| Base flow | 447660 | 601184 | 577108 | 539536 | 34.29 | 28.92 | 20.52 |
| 20-Year | 529204 | 695236 | 632712 | 590280 | 31.37 | 19.56 | 11.54 |
| 50-Year | 556816 | 757300 | 707024 | 637320 | 36.01 | 26.98 | 14.46 |
| Roughness height $k_s = 0.3$ m | | | | | | | |
| Base flow | 493028 | 635484 | 599948 | 561700 | 28.89 | 21.69 | 13.93 |
| 20-Year | 604988 | 741964 | 689388 | 636916 | 22.64 | 13.95 | 5.28 |
| 50-Year | 747764 | 898048 | 859004 | 824928 | 20.10 | 14.88 | 10.31 |

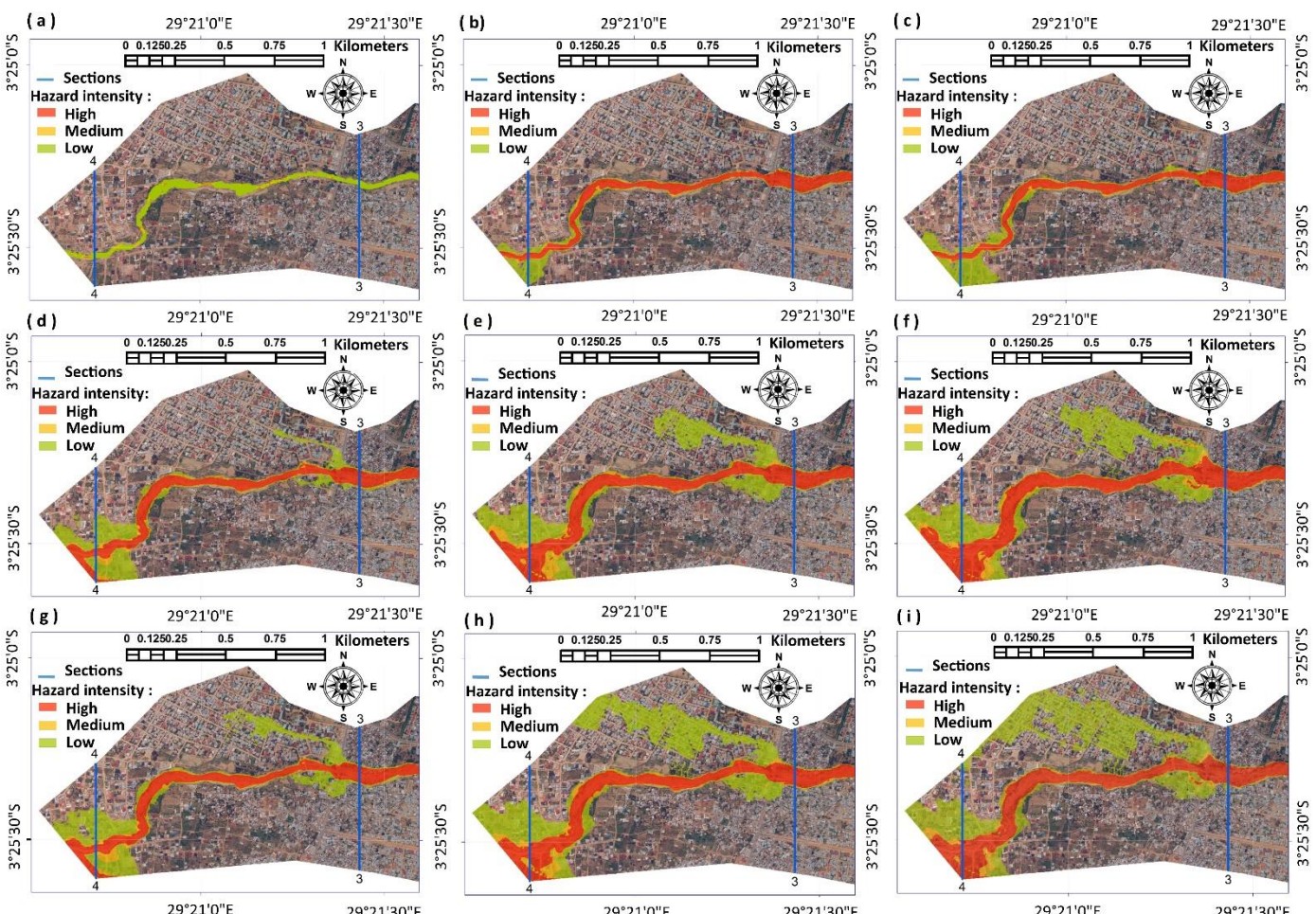

**Figure 15a.** Flood intensity maps for various initial steady discharges and roughness: the first column (a, d, g) corresponds to the pre-failure scenarios while the second (b, e, h) and third (c, f, i) columns relate to the gradual (10 minutes as breaching time) and instantaneous breaching. The first line (a, b, c) is based on the base flow and a roughness height of 0.1 m. The scenarios of the second line (d, e, f) are simulated using a 50 years-flow and a roughness of 0.1 m. The third line (g, h, i) is similar to the second one, but considers a roughness height of 0.3 m.

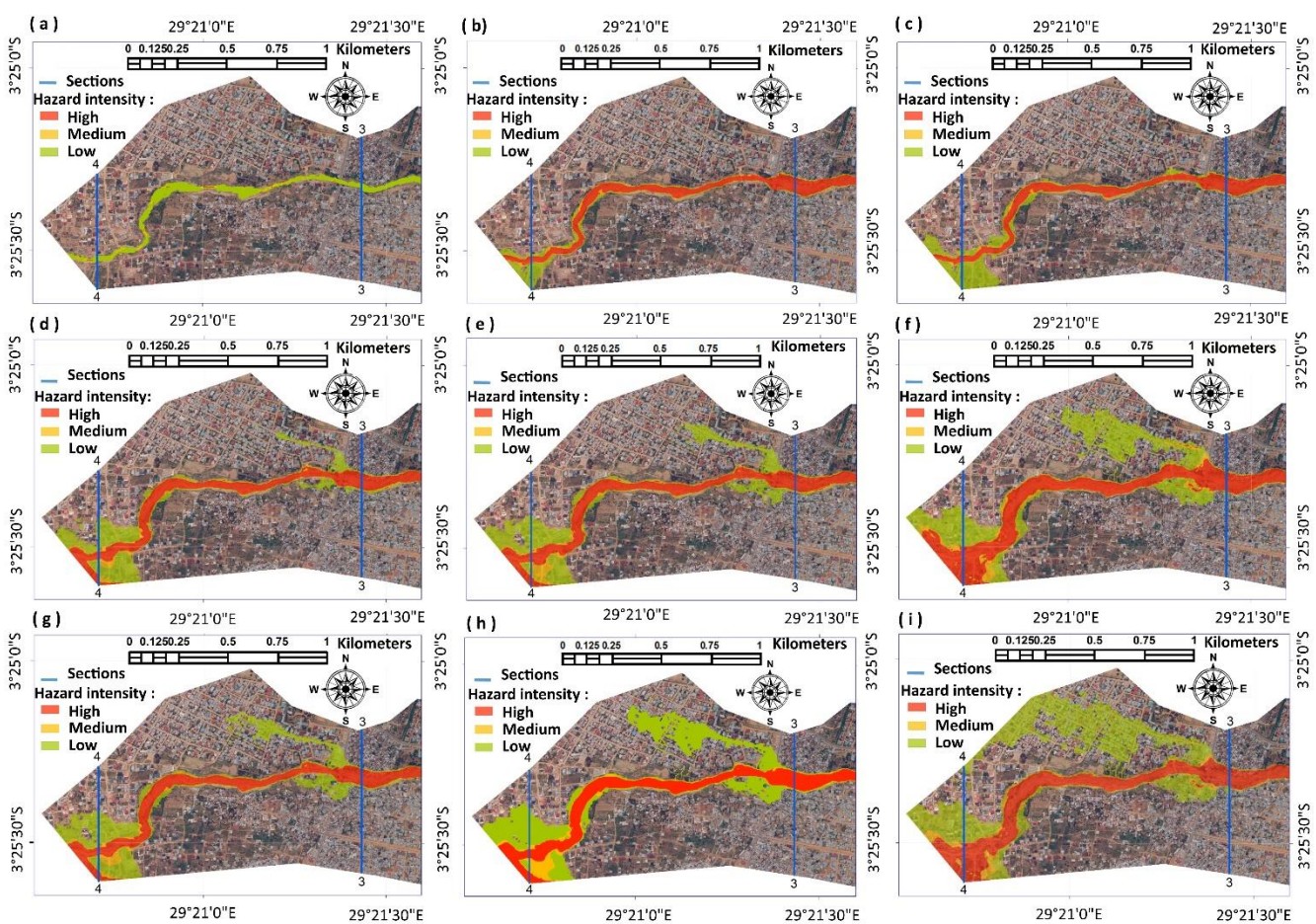

**Figure 15b.** Flood intensity maps for various initial steady discharges and roughness: the first column (a, d, g) corresponds to the pre-failure scenarios while the second (b, e, h) and third (c, f, i) columns relate to the gradual (60 minutes as breaching time) and instantaneous breaching. The first line (a, b, c) is based on the base flow and a roughness height of 0.1 m. The scenarios of the second line (d, e, f) are simulated using a 50 years-flow and a roughness of 0.1 m. The third line (g, h, i) is similar to the second one, but considers a roughness height of 0.3 m.

## 4    Discussion

### 4.1    Comments on the landslide analysis

The main question about the present state of the BTL has already been introduced above: under certain conditions BTL is likely to be destabilized, but is a full blockage of the river possible?

In addition to above modelling results, we present here some direct proofs of the likely future massive activation of the landslide – under certain conditions (similar to the simulated worst-case scenarios). First, the scenario of future formation of a landslide dam is supported by observations indicating that the landslide had already formed a dam in the past. Actually, directly upstream from the landslide the valley widens and it is filled both by coarse and by fine deposits. Especially, the latter

indicate that a lake has existed upstream form the landslide, probably due to the damming of the river that must have lasted a certain time (probably months or years). Second, many ground cracks as well as rock structures favouring sliding along the slope (Figs. 2b and 2d) were found on the landslide surface and at its foot, respectively. Apart from the fact that these cracks and layers constitute zones of weakness, they contribute to the landslide destabilization by diverting large quantities of the runoff water to the inner part of the landslide and to the main sliding surface. This water can contribute to the lubrication of the clay that may then form 'soap layers' (see such 'soap layer' surface in Fig. 2c), or by the recharge of the aquifer whose rise leads to the slope instability as shown in in the sections above. Due to the landslide surface morphology, water could accumulate at its surface and form some ponds (see view of main pond in Fig. 2a). Those ponds do not only contribute to the saturation of the soil, but they also constitute an additional active load for sliding. One scenario that could not be simulated includes the opening of fractures below those ponds that would drastically increase the groundwater pressures at depth. All these elements allow us to validate the simulated scenarios considering worst case conditions (high groundwater pressure, seismic activation) and indicate that even much larger movements could occur than those that were modelled: seismic vibrations could contribute to fracture opening, which in turn would allow rapid inflow of surface (and runoff) water, which could result in massive movements of materials. At least, a 15 m-high landslide dam could form: our simulations resulted in such a 15 m-high dam along the river axis – but which did not fully block the river section as the 12 m horizontal displacement would still allow the river to flow around the landslide; larger horizontal displacements such as those expected after pouring of all existing pond waters into the landslide, down to the sliding surface, would probably result in a full river blockage. Behind this dam a water impoundment of about $60 \times 10^3$ m³ or more could develop. For the evaluation of this volume we consider the extension of past lakes that had been dammed by the same landslide as proved by the presence of lake sediments directly upstream from the landslide (covering a surface area of about 12000 m²).

## 4.2    Key findings from the hydraulic modelling

One of the key elements highlighted by our flood scenario analysis is the influence of the surface roughness on the dynamics of the Kanyosha River. The studied dam failure scenarios complete the findings of the stationary analysis by providing a better understanding of the hydrological behaviour of the Kanyosha River. Most importantly we found that, according to the worst case scenarios, a large flow discharge is expected to arrive very quickly near the inhabited regions, which might not allow the inhabitants to escape. This result is strongly depending on the river bed roughness change, potentially due to previous floods and/or anthropogenic disturbances. These findings are of a great interest, as they can help decision makers to promote a non-risky city management near Kanyosha River and other rivers in similar conditions, by controlling all activities that can alter the roughness of the rivers, knowing their effects on the severity of flooding. Flood intensity maps are valuable tools showing the areas that can be affected under different scenarios and helping to take adequate measures to avoid losses due floods. The effects of dam failure on the flood intensity are well highlighted. Significant changes in failure scenarios computed only with base flow constitute the most important element in risk prevention. Indeed, warning systems are based on data provided by meteorological services analysing the likelihood of heavy rainfall. However, dam failures can produce floods that are several times more severe than those caused by concentrated surface runoff. This shows that dam failure can distort flood forecasts, creating surprises through non expected circumstances. Hence multi-hazard analyses remain of great interest in high geological risk environments such as those found along the East African Rift system.

## 4.3    Uncertainties and limitations

### 4.3.1.    Influence of general assumptions and parametrization

The characteristic size of the bottom irregularities was observed to vary along the river channel. Therefore, although we tested different values of the friction coefficient in our simulations, uncertainties remain regarding the effect of the spatial variability in bottom roughness.

In our simulations, we also assume that the reservoir behind the dam is completely filled when the failure starts. The actual

situation could be different, as the breaching may occur before the complete filling of the reservoir. However, in such a case, the severity of the induced flooding would be lower, so that our assumption makes sense from the perspective of risk management. Filling of the reservoir takes about 5.5 hours, 17 minutes and 9 minutes in, respectively, the base flow scenario, the 20-year flood scenario, and the 50-year flood scenario. This remains of the same order as the typical lifespan of a landslide dam.

Moreover, the dam breaching mechanism and dynamics depends on a series of factors related to the resistance of the natural dam. Although it may considerably affect the actual breaching and the induced flood wave, the detailed prediction of this resistance is out of the scope of the present study and was handled by reasonable and discussed assumptions of the breach formation time.

### 4.3.2.     Influence of the topographic and bathymetric data on the water depth and on the peak discharge

To assess the sensitivity of water depth and peak discharge to the DEM, we compared the results of simulations based on the initial 10 m × 10 m DEM and those based on the topographic field survey (Section 2.2). The results given in Fig. 16 allow the comparison of computed water depths and peak discharges in both cases, for various initial flow in the river and roughness heights of 10 cm or 30 cm.

Some significant deviations are found for the computed water depths, indicating that the values of water depths are strongly influenced by local details in the topographic data. These influences are highly variable in space. This is an expected result and, for instance, the water level would show a more limited sensitivity to the topographic details than the water depths do. The differences may result from the limited accuracy of the topographic datasets, from planform variations of the river channel since the riverbanks are not stabilized and frequently undergo changes due to erosion and anthropogenic disturbances. Changes may have occurred between the production of the 10 m × 10 m DEM (2012) and the field survey (2014-2015).

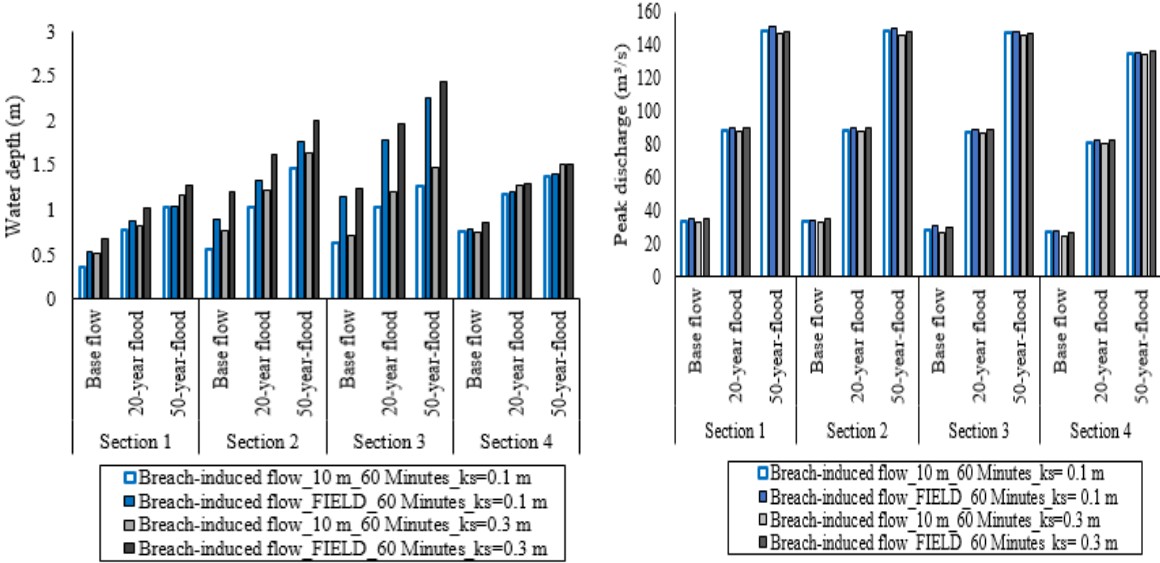

**Figure 16.** Water depth and peak discharge obtained from simulations based on the 10 m × 10 m DEM and from a topographic field survey. The breaching duration is 60 minutes.

In contrast, the differences in the peak discharges remain very limited, since they are equal to 3 % in average and they never exceed 10 % (this value is obtained for an initial base flow and a relatively high roughness height). The higher the initial flow in the river, the lower the sensitivity of the peak discharge. This suggests a reduced influence of the topographic details on the peak discharge, as also confirmed in Table S4 (Supplement 7).

To quantify the sensitivity of the flood extent to the topographic data used, an indicator was calculated based on the pixels included in the flooded area computed based on the two topographic datasets. This indicator is the ratio between the number of pixels in the intersection and the number of pixels of the union of the two computed flood extents. Its value ranges between 0 (no overlap) to 1 (perfect agreement). For $k_s = 0.1$ m, the results show an indicator of 0.82, 0.85 and 0.77 for a base flow, a 20- year flood and a 50-year flood, respectively. For $k_s = 0.3$ m, the corresponding computed indicator are equal to 0.83, 0.85 and 0.86, respectively. These results reveal a moderate sensitivity of the flood extent with respect to the two tested topographic datasets. The details of the results are provided in Table S5 (Supplement 7), considering a breaching duration of 60 min.

### 4.3.3.  Impact of solid transport on the flow

To appreciate the effect of the mobilized solid material, we used the volume of the landslide dam as a proxy for the volume of released solid material. The volume $V_d$ of the landslide dam is about 16,000 m³, while the volume $V_l$ of water impounded behind the landslide dam prior to dam breaching is roughly 55,000 m³. Table 11 provides an estimate of the ratio between the volume of dam material and the total volume of water contributing to dam erosion in the various considered scenarios. Table 11 suggests that only in the case of a 20- or a 50-year flood and a slow erosion of the dam (in hours), the volume of dam material could reasonably be neglected compared to the volume of water, as in this case, the volume of water contributing to the dam erosion is approximately twenty to thirty times larger than the volume of the dam material. In all other cases, the volume of dam material ranges between 12 % and 30 % of the water volume and is therefore not negligible.

**Table 11.** Estimated volume of water released at the dam over the breaching duration, evaluated as $V_l + T_c \times Q_r$. Notation $V_l$ refers to the volume of water initially impounded behind the landslide dam, $Q_r$ to the river discharge before dam breaching and $T_c$ is a characteristic time, taken equal to 60 s for the extreme scenario of instantaneous dam breaching and equal to $T_f$ (breaching duration) in the other cases. Notation $V_d$ designates the volume of the dam.

| Hydrological scenario | River discharge $Q_r$ before dam breaching | Dam breach scenario | | |
| --- | --- | --- | --- | --- |
| | | "Instantaneous" dam breaching | Breaching duration of 600 s | Breaching duration of 3600 s |
| Mean discharge | 3 m³/s | $5.5\ 10^4$ m³ $\approx 3.5\ V_d$ | $5.7\ 10^4$ m³ $\approx 3.6\ V_d$ | $6.6\ 10^4$ m³ $\approx 4.1\ V_d$ |
| 20-year flood | 60 m³/s | $5.8\ 10^4$ m³ $\approx 3.7\ V_d$ | $9.1\ 10^4$ m³ $\approx 5.7\ V_d$ | $2.7\ 10^5$ m³ $\approx 17\ V_d$ |
| 50-year flood | 120 m³/s | $6.2\ 10^4$ m³ $\approx 3.9\ V_d$ | $1.3\ 10^4$ m³ $\approx 8.0\ V_d$ | $4.9\ 10^5$ m³ $\approx 31\ V_d$ |

In addition, we may appreciate the plausible consequences of morphodynamic evolutions (erosion, deposition) based on the results of the sensitivity analysis conducted with respect to a change in the DEM (Subsection 4.3.2). The differences between the two considered DEMs are of course not correlated with locations of preferential erosion or deposition in the valley; but the overall order of magnitude of these differences is in agreement with a plausible amount of deposits resulting from the volume of solid material released during the breaching. Indeed, given the volume of the landslide dam ($V_d$ =16,000 m³), if we assume

an average flow width of 30 m and a sediment spread over only 1500 m, the thickness of the deposits is of the order of 35 cm. This thickness remains in the same range as the differences between the 10 m × 10 m DEM and the field measurements (Section 2.2). Therefore, we speculate that the changes in the computed flow characteristics as a result of a change of the DEM (Subsection 4.3.2) might be of the same order as those which would result from erosion and deposition of solid materials (higher sensitivity of the water depths compared to flood discharge). This requires obviously a thorough verification by means of the more sophisticated flow and morphodynamic models than used here.

## 5    Conclusions

The processes of the triggering and evolution of the Banana Tree Landslide along the slope south of the Kanyosha River near Bujumbura were analysed. A large set of simulations was computed to understand how the landslide evolved from its initial situation to the current state by back-analysis. Results showed that the sliding must have been initially triggered under extreme conditions, involving high groundwater pressures and most likely also quite strong seismic shaking. Furthermore, we showed that the Banana Tree Landslide in its present state can still lead to disasters in the future, as the combination of earthquakes and increased groundwater pressures could result in massive downslope movements.

It should be highlighted that the landslide is still active, especially within the downstream block where the river erosion at the foot of the slope and the ground saturation are accelerating sliding processes. Enhancement of those processes (by higher groundwater pressures, possibly also due to seismic shaking, and/or due to ground cracks allowing for rapid surface water infiltration, etc.) will inevitably lead to larger movements and the formation of a landslide dam, behind which a large lake could develop.

A hydraulic model provided valuable quantitative information on the flood wave characteristics and propagation resulting from a possible landslide dam breach. Here, we primarily considered the pre-condition of a total dam formation and a later (more or less) sudden and full collapse leading to a rapid release of (possibly all) the waters stored behind the dam. It enabled us to assess quantitatively different failure scenarios as well as the influence of various parameters. One of the most important conclusions of this work is that some areas assumed to be in security with respect to regular floods related to simple concentrated surface water runoff might become exposed to extreme flooding in case of an upstream dam failure. Hence, it is important to take these realities into account in a sustainable spatial management planning and especially in areas marked by high population densities. Flood intensity mapping is still a valuable tool and can be used as a guide, helping decision makers in urban planning. Since some hydraulic parameters (e.g. the water depth) are sensible to topographic data, efforts have to be made to gather suitable topographic data with high resolution, in order to minimize uncertainties in flood forecasting.

As emphasized in Subsection 4.3.3, the present study should be pursued by taking into account the volume of released solid material and applying a sediment transport and morphodynamic model, as included in more advanced debris flow / granular flow modelling tools such as presented by Mergili et al. (2012a, 2012b, 2017) or others, and adapted to channelized debris flow.

### Acknowledgements

Results presented in this paper were obtained in the frame of research funded by the Burundi government who supported the PhD studies of Mr. Leonidas Nibigira and by the Belspo (Belgian Federal Science Policy) project GeoRisCA (2012-2017): Geo-Risk in Central Africa: integrating multi-hazards and vulnerability to support risk management. Elevation and meteorological data were provided by the Geographic Institute of Burundi (IGEBU) and the '*Bureau de Centralisation Geomatique du Burundi*'. Therefore, the authors are grateful to both financial supporters and data providers.

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
