# Peer review of "Formation, breaching and flood consequences of a landslide dam near Bujumbura, Burundi"

_Natural Hazards and Earth System Sciences, 2017_

## Referee Comment (RC1) · M. Mergili (Referee) · 29 Oct 2017

Review to the manuscript „**Formation, breaching and flood consequences of a landslide dam near Bujumbura, Burundi"**

*submitted by Léonidas Nibigira et al.*

*Reviewer: Martin Mergili*

The authors present an interesting study on the possible impact of the dynamics of a landslide on flooding downstream, thereby considering the effects of breaching of a landslide dam. Much of the manuscript is well written, structured, and illustrated. Particularly the evaluation of the stability conditions of the landslide is very well described. However, coupling of landslide and flood is, in my opinion, insufficiently covered. Particularly in this context I have identified some important issues requiring improvements and therefore recommend **major revisions**. I now outline my suggestions and comments in the order of decreasing priority:

1. One of my major concerns relates to the fact that only flooding by water is considered. Breach of the landslide dam would release a huge amount of solid material (most probably deeply weathered tropical soil) that would be incorporated in the flow and could possibly lead to completely different characteristics and downstream impact of flooding, compared to clear water flow. This issue is not even discussed at all. I see two possibilities to face this challenge: (i) incorporating sediment load in the flow simulation; or (ii) a thorough argumentation and discussion why this is not necessary. Either (i) or (ii) should be an absolute requirement for the acceptance of the manuscript.

2. I do not fully understand the work flow of the flood modelling: in the first step, do you (i) simulate the base flow without the dam incorporated, or do you (ii) fill the lake behind the dam to let it flow out in the second step? The description in Sect. 2.4.3 is confusing and has to be improved.

3. A highly critical issue is also the consideration of dam breach (lowering of the dam crest and release of the impounded water) – how does this work? Please explain! I have the feeling that you spend a lot of effort in describing base flow and lower boundary conditions at a high level of detail, but do not explain some of the really important aspects at all.

4. You claim to consider flood hazard – however, it is only the possible intensity which is used for the preparation of the maps – hazard would also have to include a measure for frequency. Some rewording (e.g. flood intensity indication map?) will be necessary.

5. You resample the 10x10 m DEM to 2x2 m. it is absolutely clear to me that this is necessary for numerical reasons – still, it does not increase the level of topographic detail. How wide is the river, i.e. is an effective 10x10 m cell size sufficient to capture the topographic patterns governing the flow? Please discuss.

6. The discussion on the uncertainties involved is very short, given that the uncertainty issue is very important when it comes to computer simulations. Some further considerations would be desirable (geotechnical parameters, hydrograph, sediment transport, ...)

7. Flow depth x velocity does not result in $m/s^2$, but in $m^2/s$.

I am looking forward to see a revised version of this manuscript. The authors should feel free to contact me at martin.mergili@univie.ac.at if they disagree with my comments or in case they would like to discuss the one or the other issue.

With best regards, Martin Mergili

---

## Referee Comment (RC2) · O. Dewitte (Referee) · 24 Nov 2017

[referee-annotated manuscript omitted]

---

## Author Comment (AC1) · 30 Jan 2018

**Authors response to Review comment by Olivier Dewitte (Referee) for "Formation, breaching and flood consequences of a landslide dam near Bujumbura, Burundi"**

Léonidas Nibigira[1], Hans-Balder Havenith[1], Pierre Archambeau[2], and Benjamin Dewals[2]

[1]Geohazards and Environment, Department of Geology, University of Liege, 4000 Liege-Belgium

[2]Hydraulics in Environmental and Civil Engineering (HECE), Research unit Urban & Environmental Engineering, University of Liege, 4000 Liege-Belgium

Correspondence to: Léonidas Nibigira (leonidas.nibigira@doct.ulg.ac.be)

Throughout this document, each Reviewer comment is bordered in black and is directly followed by authors' response. Sometimes, in case two or more comments have a certain affinity, they are bordered together and answered once and for all. When a rewording is proposed, the text is in italics and in quotation marks.

**1. General comment:**

Dear colleagues,

The research proposed by Nibigira and co-authors focuses on landslide damming and possible subsequent flood occurrence in Bujumbura. More specifically, the authors focus on a large landslide that develops in a watershed flowing toward the city center. The first part deals with geophysics and landslide 3D and 2 D reconstruction. The second part is with the modelling of the landslide. The third part deals with the simulations of the floods and their associated hazard induced by dam breaching. The manuscript presents interesting results for such an understudied region; especially when one considers that in these African regions data scarcity is commonplace. I agree with the comments and suggestions by the first reviewer. I have therefore done my review accordingly (see supplement material). This includes minor to moderate comments and technical corrections. Two points I insist to are the weathering of the lithology and, for the introduction, the data-scare context and methodological challenges of the study area.

Regards

Olivier Dewitte

Thank you for your very interesting and enriching comments and suggestions to our manuscript.

 **Specific comments:**

2. 1: (Line 20) The introduction can be improved as it suffers from a lack of state of the art referencing to the general literature on landslide dam and related flood modelling. As it stands now we miss the broader context of this study (scientific and societal needs to perform such a study, challenges to get relevant input data for
35  modelling, context of data scarcity in the studied region, methodological challenges, etc.). Then, based on this context, comes the objectives that you aim at studying a specific landslide in a specific region.

A rewording of the introduction is proposed below (Fig. 3 is the former Fig. 14 moved as suggested later within the comment 2.28):

[revised manuscript text omitted]

100 **Figure 2.** *(a) View of BTL (black dotted contour) and of the main scarp (red dotted contour) from downstream (other orange and light blue dotted contours mark other instabilities). The landslide sliding direction and the river flow direction are indicated by red and black arrows, respectively. AB indicates the height (=26 m) of the landslide frontal part near the river; BC outlines the BTL length in the sliding direction (~750 m); CD shows the height of the main scarp (~ 75m) along profile BCE. The blue line indicates the river channel axis. (b) Waterfall over a destroyed old hydraulic structure. (c) View of the river bed during the dry season with presence of cobbles and fine boulders that are* 105 *deposited after floods during the wet season.*

[Figure]

*Figure 3. Field observations highlighting the critical stability state of BTL. (a) View of the rock structures at the foot of the landslide (note the boulder marked by a red cross in the lower left part has a diameter of about 60 cm), generally dipping towards the north (left side), parallel to the sliding direction (red arrow). The blue arrow indicates the river flow direction. (b) View of a crack on the sliding interface in clay. The red arrow shows the direction of sliding of the right part along the clay layer. (c) Pond on the landslide with an oil palm designated by the white arrow. This shows that these ponds are recent (oil palm trees do not grow in water; its particular foliage compared to others shows that its growth was stopped recently). (d) A crack found on the landslide surface.*

2. 2: (Fig. 1) The relief is represented with shiny colors. I would have combined them with a hill shade view to improve readability.

To take into account your very relevant recommendations, the map has been modified (see Fig. 1 above).

2. 3: (Fig. 1) Delete the "_"

The "_" is deleted from the legend of Fig. 1.

2.4: (Fig. 1) Color issues: blue for watershed delineation and red for a river is weird. I suggest blue for the river and another color for the watershed.

Your suggestions are taken into account: blue is used for the river network while red is used for the watershed delineation (Fig. 1).

2.5: (Fig. 2) where is the waterfall located?

This waterfall formed on a former flood control structure is located at 270 m downstream the cross section 3 shown in Fig. 4 (shown below).

2.6: (Line 50) This means that the dotted contour line used in Figure 2 represents only part of the landslide; i.e. the landslide section that you consider in the modelling. That something that should be stressed better within the Figure caption. So far, this figure shows some landslide parts such as the one where the compass rose is positioned that are not delineated alone. Based on the figure and its caption alone, it looks strange.

Your opinion is very relevant, since other instabilities appear on the figure.

Indeed, the whole site of the BTL is strongly weakened, creating small secondary landslides juxtaposed with the main landslide. These are located at the foot of the slope along the Kanyosha River and do not follow the dynamics of the global movement. They are detached by small blocks that are then subject to erosion and are therefore not likely to produce significant effects on the risk of flooding. This is well expressed from line 48 to line 51 of the manuscript.

In our opinion, the confusion comes partly from the fact that the figure is not strongly announced, leading the reader to ask himself some questions before having a look at the following parts of the manuscript. For this reason, we believe that moving the paragraph from Line 48 to Line 51 of the manuscript ("Our recent observations ... move and form a landslide dam") and inserting it before Fig. 2 of the manuscript would help to improve the manuscript, as it will serve as introduction to the Fig. 2.

For this purpose, we propose, on the agreement of the Editor, the displacement of the aforementioned paragraph in the revised version of the manuscript.

This is, with a slight correction ("*yellow and blue dotted contour*" is added in parentheses after "local *slope instabilities*"), the paragraph that will be moved to serve as introduction to Fig. 2:

*Our recent observations show that the western part of the landslide (in the foreground of Fig. 2a), with relatively soft slopes, is marked by very local slope instabilities (yellow and light blue dotted contours) that do not contribute to the general movement. However, the eastern part (black dot outlines in Fig. 2a) presents steep slopes near the river; this active zone is 250 m wide and could soon move to form a landslide dam.*

Moreover, related information will be added within the Fig. 2 caption as follows for much more details:

*Figure 2. (a) View of BTL (black dotted contour) and of the main scarp (red dotted contour) from downstream (other orange and light blue dotted contours mark other instabilities). The landslide sliding direction and the river flow direction are indicated by red and black arrows, respectively. AB indicates the height (=26 m) of the landslide frontal part near the river; BC outlines the BTL length in the sliding direction (~750 m); CD shows the height of the main scarp (~ 75m) along profile BCE. The blue line indicates the river channel axis. (b) Waterfall*

155 *over a destroyed old hydraulic structure. (c) View of the river bed during the dry season with presence of large boulders that are deposited after floods during the wet season.*
* * *
2.7: (Line 51) The presence of water ponds is a sign of the landslide current activity. However, can you elaborate more on the role of those ponds on future instability? Do you have references to support this?
* * *
First, this idea is strongly supported later by our results. In fact, these ponds contribute to the saturation of the
160 landslide body, whereas in sections 3.1 and 3.2, we have quantified the impact of groundwater on landslide, as it greatly reduces the factor of safety (FoS) and increases X-acceleration.

The weight of water ponds also contributes to the loading of the sliding. The impact of the slope overloading on the landslide dynamics is also strongly highlighted both by Terzaghi (1950), Varnes (1978), Popescu (1994) and Popescu (2002).
* * *
165 2.8: (Line 53) well, I wonder how one single landslide could have on impact in the country's economy. I suggest to rephrase this in a more balanced way: "... makes it a potential danger for people and infrastructures of that area".
* * *
Yes, a rewording of Line 53 of the manuscript will be made to include your suggestion. This was the result of an error in writing and the correction is already included at Line 74 of this document, within the proposed rewording
170 of the introduction in our reply to comment 2.1 above. Our intention was to apply "imminent danger" only to "for life" and point out the possibility of slight economic issue, regardless of the degree of impact. Based on the disastrous consequences of the landslide along the Gasenyi River in February 2014, there is a real disruption at least for the short term. More than 940 homes destroyed and 12500 homeless people to manage, especially in a non-preparedness situation. Reconstruction and studies as well as corollary changes to operate to avoid new
175 disasters in the area etc., can require a real effort in the context of a fragile economy.

Moreover, the BTL is more than 3 times larger than that of the Gasenyi River (under 10 m high for the Gasenyi landslide dam while the BTL can potentially be 15 m-20 m high).

To give an idea of what such disastrous event can coast to the economy, we would like to present a brief analysis of the flood of 2014.

180 First, it should be noted that for 2014, the Government of Burundi adopted an austerity budget policy (that was called '**GUTUBIKA UWAVUBI**' in Kirundi, the Burundian language), with an annual budget of 1403.3 billion Burundian Francs (909.8 million USD).

Following the catastrophic floods that occurred in February 2014 (Table 1), a joint mission of some agencies and programs of the United Nations, the European Union, the African Development Bank and the World Bank was
185 deployed to evaluate the disaster.

Overall, according to estimates given by the technical teams, the losses related to public infrastructure, Crops and private houses exceeded 18.9 Million USD. That is about 2.1% (of which 0.49% for the public infrastructures and 1.6% for Crops and private houses) of the annual budget of the State.

However, the rehabilitation and management measures and especially the resulting disaster prevention measures are much cheaper than the initial damages. These fees were estimated at 105.7 million USD, the equivalent of 11.61% of the annual budget. This is, unfortunately, a real challenge, in the economic context mentioned above.

In terms of harvests, an estimate was made by FAO while for infrastructure, the evaluation was carried out by a team of UNDP and IOM, supported by both the Road Sector Project Implementation Units and the Public Works & Urban Management unit.

(FAO: Food and Agriculture Organization, UNDP: United Nations Development Program)

> 2.9: (Line 62: channel description) I know that they are some definitions related to the size. I am not sure a 40 cm diameter is appropriate for a pebble. Not even 10 cm to be checked with a classification system adapted for your study.

The extended Udden-Wentworth grain-size scale nomenclature will be applied, as proposed later in our reply for the comment 2.12. Grain-size in the text proposed to comment 2.12 is reworded accordingly.

> 2.10: (Line 63: channel description) size of those stones? Usually, above few tens of cm we call them boulders.

A rewording is proposed within the reply to 2.12.

> 2.11: (Line 65: channel description) Accumulation zones of what?

This section is improved by the use of a suitable grain-size classification. The accumulation consists of fine materials (clay and silt) trapped in small flats or generally upstream of the remains of old hydraulic structures. Often, within these areas develop small green islets like those shown in Fig. 2.b. The rewording proposed in the reply to the comment 2.12 includes all these details.

> 2.12: (Line 65: channel description) Do you make a distinction between pebbles and debris? If so, where does debris come from? More info needed.

Regarding the comments from 2.7 to 2.10 and their reply given above, a rewording of the channel description (Sec. 2.1) is proposed below:

*"The Kanyosha River main channel has deposits with variable grain size. Based on the extended Udden-Wentworth grain-size scale nomenclature (Terry and Goff, 2013), the riverbed material can be classified into three main groups. The first consists of cobbles of around 10 cm in diameter or more (Fig. 2c). The coarse part of this category consists of fine boulders, with a diameter generally under 40 cm. The second group is made up of isolated medium boulders that are often prone to the action of humans, carving them into building materials (mainly paving plates).*

*This category is difficult to take into account due to its strong irregularity. The third group consists of silt and clay zones, generally near former hydraulic structures in the downstream part of the river. In this category, we can mention small*
220 *herbaceous islets, often located near the river overbanks. As in the second group, this category is found only in small isolated and scattered areas, subject to strong seasonal variations. Globally, the first group remains hydraulically predominant. Here, the variability of the grain size was accounted for by means of sensitivity analysis (Sec. 3.3).*

*In 2006, hydraulic structures were constructed to regulate the river; but they were quickly damaged by floods during the following raining seasons. Nonetheless, isolated cobbles resulting from the destruction of these structures are observed. They*
225 *join the second group described above. The accumulation of material upstream of the remains of the structures often form horizontal platforms, generating small waterfalls (Fig. 2b)."*

Details for the Udden-Wentworth grain-size scale nomenclature are provided in Table 1 below:

| PARTICLE LENGTH (dI) km | m | mm | Φ | GRADE | CLASS | FRACTION Unlithified | Lithified |
|---|---|---|---|---|---|---|---|
| 1075 | | | -30 | very coarse | Megalith | Megagravel | Mega-Conglomerate |
| 538 | | | -29 | coarse | | | |
| 269 | | | -28 | medium | | | |
| 134 | | | -27 | fine | | | |
| 67.2 | | | -26 | very fine | | | |
| 33.6 | | | -25 | very coarse | Monolith | | |
| 16.8 | | | -24 | coarse | | | |
| 8.4 | | | -23 | medium | | | |
| 4.2 | | | -22 | fine | | | |
| 2.1 | | | -21 | very fine | | | |
| 1.0 | 1048.6 | | -20 | very coarse | Slab | | |
| 0.5 | 524.3 | | -19 | coarse | | | |
| 0.26 | 262.1 | | -18 | medium | | | |
| | 131.1 | | -17 | fine | | | |
| | 65.5 | | -16 | very coarse | Block | | |
| | 32.8 | | -15 | coarse | | | |
| | 16.4 | | -14 | medium | | | |
| | 8.2 | | -13 | fine | | | |
| | 4.1 | 4096 | -12 | very coarse | Boulder | Gravel | Conglomerate |
| | 2.0 | 2048 | -11 | coarse | | | |
| | 1.0 | 1024 | -10 | medium | | | |
| | 0.5 | 512 | -9 | fine | | | |
| | 0.25 | 256 | -8 | | | | |
| | | 128 | -7 | coarse | Cobble | | |
| | | 64 | -6 | fine | | | |
| | | 32 | -5 | very coarse | Pebble | | |
| | | 16 | -4 | coarse | | | |
| | | 8 | -3 | medium | | | |
| | | 4 | -2 | fine | | | |
| | | 2 | -1 | | Granule | | |
| | | 1 | 0 | very coarse | Sand | Sand | Sandstone |
| | | 0.50 | 1 | coarse | | | |
| | | 0.25 | 2 | medium | | | |
| | | 0.125 | 3 | fine | | | |
| | | 0.063 | 4 | very fine | | | |
| | | 0.031 | 5 | coarse | Silt | Mud | Mudstone or Shale |
| | | 0.015 | 6 | medium | | | |
| | | 0.008 | 7 | fine | | | |
| | | 0.004 | 8 | very fine | | | |
| | | 0.002 | 9 | | Clay | | |
| | | 0.001 | 10 | | | | |
| | | 0.0005 | 11 | | | | |
| | | 0.0002 | 12 | | | | |
| | | 0.0001 | 13 | | | | |

Table 1: Extended Udden-Wentworth grain-size scale for sedimentary particles, after Blair and McPherson (1999). (Source: Terry and Goff,
230 2013).

2.13: (Fig. 3) Colors are not used at their best. Suggestions for improvement: for elevation: avoid blue. As you are studying flood, it is pretty confusing. If you do not use blue for topography, you can use it for the river (which sounds more logical). Then the cross sections can be in black.

2.14: (Fig. 3_ caption) As it reads now, it is pretty confusing. It seems that you have used a hillshade product for the modelling. In addition, the figure shows a mix of DEM and hillshade.

2.15: (Fig. 3_caption) No need to repeat in the caption what you have in the legend.

Here bellow is the proposition for Fig.3 (according to the proposition, the number of that figure will be 4 instead of 3, due to the previous insertion of the former Fig.14) and its caption, based on comments 2.13, 2.14 and 2.15:

[Figure]

**Figure 4.** Digital elevation model (m) used for hydraulic modelling within the computation domain with cross sections where hydrographs were extracted and dam location. The river main channel is also highlighted.

 Up to now, there is no reference in the landslide description that the material is made of rock. From figure 2, it looks as if most of the landslide material is highly weathered. I think it is important to mention this weathering issue as this is something very specific of this type of tropical climate.

The original material of BTL is a gneiss which, by the alteration, is partially transformed into a clay on the surface. The depth of the altered layer is about 20 m.

The study area experiences alternations between dry and rainy seasons. The long dry season (from June to September) is followed by the small rainy season (from October to December), then by the small dry season (during the months of January and February). The cycle ends with the strong rainy season from March to May, just before the return of the dry season. Since the photos in Fig. 2 were taken in October, the ground was relatively wet, but not quite enough compared to December and the strong rainy season.

Especially for the low parts of the landslide, the humidity is never very low due to the recharge of the water table by the ponds of water located on the landslide. On the other hand, the groundwater recharge follows the dynamics of the seasons. In the context mentioned above, the action of the rainy season in the body of the landslide is quickly sensible, due to the water that sneaks in the interstices.

2.17: (Line 96) GPS? Why not deriving the profile from the DEM?

The geophysical survey technique by electric tomography profiles requires elevation values at each of the electrodes positioned every 5m. Thus, additional elevation values were needed to complete the 10X10 m DEM.

2.18: (Line 110) Can you say something about the material and the weathering conditions?

Indeed, the details given in response to comment 2.16 above in terms of material and weathering conditions will be inserted in the manuscript, Line 110, just after the sentence "It corresponds to the average type of the material found within the landslide". The text should be the following:

"*The original material of BTL is a gneiss which, by the alteration, is partially transformed into a clay on the surface. The depth of the altered layer is about 20 m. The study area experiences alternations between dry and rainy seasons. The long dry season (from June to September) is followed by the small rainy season (from October to December), then by the small dry season (during the months of January and February). The cycle ends with the strong rainy season from March to May, just before the return of the dry season. Since the photos in Fig. 2 were taken in October, the ground was relatively wet, but not quite enough compared to December and the strong rainy season. Especially for the lower parts of the landslide, the humidity is never very low due to the recharge of the water table by the ponds of water located on the landslide. On the other hand, the groundwater recharge follows the dynamics of the seasons. In the context mentioned above, the action of the rainy season in the body of the landslide is quickly sensible, due to the water that sneaks in the interstices.*".

275

2.19: (Fig. 6) Not visible (The point 14)

2.20: (Fig. 6) X scale of Figure 5 starts with 0 close to the landslide main scarp and the river axis is at 900 m...
I suggest the same scale is used here for the sake of clarity.

The modified version of Fig.6 (the number of that figure will be 7 instead of 6 for reasons mentioned above) is
proposed below:

280

[Figure]

**Figure 7.** Materialization of blocks, joints and materials for the actual model. The history (measurement) points 12, 13 and 15 (white dots) located, respectively, on the upper, middle and lower block correspond to the surface area where parameters were monitored (e.g. the $x$-acceleration). The point 14 is located at the basis of the model, within the bedrock. The axis of the Kanyosha River is located to the right of the history point 15.

285

2.21: (Line 140) References that justify these options? Can you discuss this?

Even tough each case is different, data availability helps to refer to an existing data base and case studies within
the study area. Unfortunately, considering the context of data scarcity in the region, it is not easy to find related
references.  This is why we did not assign a single value, but 4 different shaking duration values to well illustrate
the behavior of the model corresponding to different scenarios. The seismic context is analyzed on the basis of

290

earthquakes data from the Global Seismographic Network stations of the Incorporated Research Institutions for Seismology (IRIS) on the Lake Tanganyika Region. Therefore, based on that situation and local site effects, the 0.1g-wavelet used is chosen as reasonable value to predict the behavior of the landslide under very moderate seismic shaking.

295

2.22: (Line 264) Replace by "could".

This will be corrected in the revised version of the manuscript.

2.23: (Line 265) The origin of the water pounds present in the landslide can also be of runoff concentration origin.

Yes, we do agree with you and this was already mentioned in the discussion section within the first version of the manuscript (Line 399 of the manuscript).

300

2.24: (Line 265) You mean, in the landslide body?

We mean water ponds on the landslide (BTL), as shown in Fig. 3c above (Fig. 14c in the manuscript).

2.25: (Line 269) You mean BTL landslide is it?

Yes. To avoid any confusion, we shall change "Kanyosha Landslide" into "BTL".

2.26: (Line 271) Have you observed/measured this, or are they hypothesis. For such a large landslide, the response

305 of the displaced material can be more complex that just following the seasonality of the wet periods.

This statement is not limited to the BTL, but rather most landslides in Bujumbura, especially those located along the Mugere River (in the south of the City Bujumbura) and Muha River (north of the Kanyosha River). From 2012 to 2016, field investigations were carried out to assess the risk of landslides. Benchmarks had been installed at some specific scarps and cracks' boundaries. Their relative position change and distances were measured over

310 time to follow the movement of the ground. Therefore, this is based on field observations and measurements.

2.27: (Line 397) Nowhere earlier clay is mentioned as being a component of the landslide material. Some info needs to be provided earlier.

We think this is now corrected and it is compliant with the details added above in the material description. This is also corrected by moving the Fig. 6 (which also includes a clay description and its role in the landslide dynamics)

315 as suggested within the comment 2.28 below.

2.28: (Fig. 14) This figure must appear much earlier in the text, when description of the landslide is made. This figure does not provide results s.s. and in addition is called just after Figure 2.

This figure will be moved and inserted in the introduction (section 1), as it is within the reworded text of the introduction proposed above.

320

2.29: (Fig. 14_Caption) Is this important to focus on the river bed material in this context?

It is not very important, as the main information here is the orientation of layers.

2.30: (Line 436) To add a comment to those of reviewer 1, discussion should also focus on the extension of the hazard zone with regard to the use of a 10 m resolution DEM. In addition, being familiar with the region, I know that river banks can sometimes be of several meters high, i.e. well above the water depth that you modelled. Can
325 you comment on that as well?

The reply to the question on the extension of the hazard zone will be included in the reply to the Reviewer 1 Comments (2.6).

Regarding the compatibility of our results with the field reality in terms of water depth, we strongly agree with you that the river banks can sometimes be several meters deeper.

330 Our results are rather satisfactory and are in line with that statement for the following reasons:

    i.      Water depth as given in Figure 11a and 11b of the manuscript show that, for scenarios leading to the water overflow beyond the main channel (e.g. breach-induced flow), the water depth values at the cross section reach 3.5 m - 4 m in the upstream parts (cross section 1 and 2) and 1.8 m – 2 m downstream (cross section 3 and 4). This corresponds well to the field reality and is in line with the
335                 digital terrain model. In the urban area, the main channel depth is not very high and, at some locations, a water depth greater than 1.5 m can lead to a considerable floodplain width.

    ii.     Considering the entire modelled river reach, the water depth varies from 0 m to 15.95 m. That means that there are already modelled values greater than those presented at the cross sections. As this can be well visualized in the Fig. R1 below, cross sections do not correspond to the maximum modelled
340                 water depth.

[Figure]

**Figure R1**. View of water depth map between cross sections 1 and 2, corresponding to the breaching-scenario, with a 50 year-flood.

This means that high values do exist, but they are not necessarily localized in cross-sections, as the cross sections' location was determined before the model run while water depth and other hydraulic characteristics are provided in the model output.

iii.    Due to the asymmetry of some cross sections, the river overflows on the side corresponding to the lower overbank. Thus, the water depth may not always reach the level of the highest river banks.

**References**

Adams, J.E.: Earthquake-dammed lakes in New Zealand, Geology, 9, 215-219, 1981.

[revised manuscript text omitted]

---

## Author Comment (AC2) · 2 Feb 2018

**Author response to Review comment by Martin Mergili for "Formation, breaching and flood consequences of a landslide dam near Bujumbura, Burundi"**

Léonidas Nibigira[1], Hans-Balder Havenith[1], Pierre Archambeau[2], and Benjamin Dewals[2]

[1]Geohazards and Environment, Department of Geology, University of Liege, 4000 Liege-Belgium

[2]Hydraulics in Environmental and Civil Engineering (HECE), Research unit Urban & Environmental Engineering, University of Liege, 4000 Liege-Belgium

Correspondence to: Léonidas Nibigira (leonidas.nibigira@doct.ulg.ac.be)

In this document, the Reviewer comments are shown in the boxes and are directly followed by the authors' response.

**General comment**

> The authors present an interesting study on the possible impact of the dynamics of a landslide on flooding downstream, thereby considering the effects of breaching of a landslide dam. Much of the manuscript is well written, structured, and illustrated. Particularly the evaluation of the stability conditions of the landslide is very well described. However, coupling of landslide and flood is, in my opinion, insufficiently covered. Particularly in this context I have identified some important issues requiring improvements and therefore recommend **major revisions**. I now outline my suggestions and comments in the order of decreasing priority:

Thank you for the attention paid to our manuscript.

**Specific comments**

> 2.1. One of my major concerns relates to the fact that only flooding by water is considered. Breach of the landslide dam would release a huge amount of solid material (most probably deeply weathered tropical soil) that would be incorporated in the flow and could possibly lead to completely different characteristics and downstream impact of flooding, compared to clear water flow. This issue is not even discussed at all. I see two possibilities to face this challenge: (i) incorporating sediment load in the flow simulation; or (ii) a thorough argumentation and discussion why this is not necessary. Either (i) or (ii) should be an absolute requirement for the acceptance of the manuscript.

We indeed only included water in the flood wave computation, while breaching of the landslide will release a substantial amount of solid material. The resulting flow will have an intermediate behavior between clear-water flow and debris or granular flow. This issue will be acknowledged explicitly in the revised manuscript, and its implications will be discussed in detail, as proposed hereafter.

**Table R1.** Some recent studies of flooding induced by the breaching of landslide dams, and of debris flow routing.

| | Model dimensions | Morphodynamics | Flow rheology | Available observations |
|---|---|---|---|---|
| Present study | 2D | No | Clear water (turbulent flow) | None |
| Fan et al. (2012) | 1D for river flow, 2D for overland flow | No | Turbulent flow | Peak discharge, peak arrival time … |
| Yang et al. (2013) | Sobek-1D and -2D | No | Turbulent flow | Flooding occurrences |
| Shrestha and Nakagawa (2016) | 1D for river flow | Yes | Granular, hyper-concentrated and turbulent flow | Observed flood discharge |
| Li et al. (2011) | 1D for river flow, 2D sediment transport | Yes | Empirical equations for Mohr-Coulomb, viscous and turbulent shear stresses | Downstream hydrograph, observed sediment depths … |
| Mergili et al. (2012a) [NHESS] | 2D, considering bottom curvature and steep slope effects | Deposition of granular material represented explicitly | Granular flow (Savage-Hutter type model) | Focused on avalanche flows, not flooding due to dam breaching |
| Mergili et al. (2012b) [Nat. Hazards] | 2D | Sediment detachment by runoff and routing of debris flow | Semi-deterministic two-parameter friction model | Debris flow travel distance, shape of deposits … |

"As summarized in Table R1, some recent studies neglected sediment transport in the analysis of floods induced by the breaching of landslide dams (Fan et al., 2012; Yang et al., 2013), while others did take sediment transport into account (Li et al., 2011; Shrestha and Nakagawa, 2016). Indeed, sediment transport may have considerable implications on the volume of mobilized material as well as on morphodynamic evolutions of the valley bottom (e.g., sediment deposition). Nonetheless, in the particular context of the present study, going for more complexity in the modelling framework (i.e. including sediment transport) would not substantially reduce the overall level of uncertainty mainly because validation data are neither available for our case study nor for any similar one in the region, which remains largely understudied. Table R1 shows that previous studies which considered sediment transport benefited from available validation data, such as observed flood discharges or depths of sediment deposits."

In the revised manuscript, we will additionally handle the issue of sediment transport through a comprehensive sensitivity analysis, aiming at appreciating successively the effect of *(i)* the volume of mobilized material and *(ii)* the consequences of morphodynamic evolutions (erosion, deposition).

*Effect of volume involved in the flow*

The volume $V_d$ of the landslide dam is about 16,000 m³, while the volume $V_l$ of water impounded behind the landslide dam prior to dam breaching is roughly 55,000 m³. Table R2 provides a estimate of the ratio between the volume of dam material and the total volume of water contributing to dam erosion in the various considered scenarios. Table R2 suggests that only in the case of a 20- or a 50-year flood and a slow erosion of the dam (in hours), the volume of dam material could reasonably be neglected compared to the volume of water, as in this case,

the volume of water contributing to the dam erosion is approximately twenty to thirty times larger than the volume of the dam material. In all other cases, the volume of dam material ranges between 12 % and 30 % of the water volume and is therefore not negligible.

55    We propose to address this in the revised manuscript by conducting additional simulations in which the dam material is assumed "fluidized" as the breaching develops, instead of being "removed" from the simulations as it is the case now (in accordance with common practice in risk analysis of engineered dams). This will provide some hints on the influence of the overall volume of material (although assumed fluid) involved in the flow.

60    **Table R2.** Estimated volume of water released at the dam over the breaching duration, evaluated as $V_l + T_c \times Q_r$. Notation $V_l$ refers to the volume of water initially impounded behind the landslide dam, $Q_r$ to the river discharge before dam breaching and $T_c$ is a characteristic time, taken equal to 60 s for the extreme scenario of instantaneous dam breaching and equal to $T_f$ (breaching duration) in the other cases. Notation $V_d$ designates the volume of the dam.

| Hydrological scenario | River discharge $Q_r$ before dam breaching | Dam breach scenario | | |
|---|---|---|---|---|
| | | "Instantaneous" dam breaching | Breaching duration of 600 s | Breaching duration of 3600 s |
| Mean discharge | 3 m³/s | $5.5 \ 10^4$ m³ ≈ 3.5 $V_d$ | $5.7 \ 10^4$ m³ ≈ 3.6 $V_d$ | $6.6 \ 10^4$ m³ ≈ 4.1 $V_d$ |
| 20-year flood | 60 m³/s | $5.8 \ 10^4$ m³ ≈ 3.7 $V_d$ | $9.1 \ 10^4$ m³ ≈ 5.7 $V_d$ | $2.7 \ 10^5$ m³ ≈ 17 $V_d$ |
| 50-year flood | 120 m³/s | $6.2 \ 10^4$ m³ ≈ 3.9 $V_d$ | $1.3 \ 10^4$ m³ ≈ 8.0 $V_d$ | $4.9 \ 10^5$ m³ ≈ 31 $V_d$ |

65    *Effect of morphodynamic evolutions*

In our reply to comment 2.5, we propose to report in the revised manuscript on an additional set of simulations to test the sensitivity of the modelling results to the use of a different DEM (derived from field survey). This additional simulation will also give some insights into the effect of changes in the river bathymetry (e.g. as could be obtained as a result of erosion / deposition, which will not be modelled explicitly) and we suggest to link this to the present

70    comment. We may also consider running additional simulations in which we include changes in the DEM to mimic plausible deposits in the downstream (e.g., where the longitudinal slope decreases sharply). The results of these additional simulations will enable appreciating the influence of possible deposits on flooding.

Thanks to the sensitivity analysis conducted based on the proposed additional model runs, we will assess in the

75    revised manuscript which parts of our conclusions are strong despite the existing uncertainties and which ones are more affected by the modelling uncertainties linked to sediment transport. In addition, we will clearly indicate as a perspective in the Conclusion section of the revised manuscript that the present study should be further continued using more advanced debris flow / granular flow modelling tools such as presented by Mergili et al. (2012a, 2012b, 2017) or others, and adapted to channelized debris flow.

80

> 2.2. I do not fully understand the work flow of the flood modelling: in the first step, do you (i) simulate the base flow without the dam incorporated, or do you (ii) fill the lake behind the dam to let it flow out in the second step? The description in Sect. 2.4.3 is confusing and has to be improved.

In the first step, we fill the lake behind the dam to let it flow in the second step. This will be clarified in the revised
85 manuscript, by introducing a new table (Table R3) and by rewording section 2.4.3 as detailed hereafter:

"The hydraulic simulations aim at evaluating the impact of the dam failure as a result of the water impoundment behind it and the river overflowing the dam crest. Thus, the initial step of hydraulic modeling considers a filled reservoir and a steady flow of water over the crest of the dam before failure. In line with Dewals et al. (2011), the modelling procedure involves two steps:

90 - step 1: a pre-failure steady flow is computed in the river, under three different hydrological scenarios (steady flow corresponding to the mean discharge in the river or to a 20-year flood, or a 50-year flood);
- step 2: using the result of step 1 as initial condition, the flow induced by the breaching of the dam is computed.

In Step 1, the dam geometry is incorporated in the topographic data used for flow computation. This means that
95 the dynamics of material sliding into the river is not explicitly reproduced in the hydraulic modelling. As it is not possible to anticipate when the landslide dam breaching might occur, we consider three different pre-failure flow conditions: base flow, 20-year flood and 50-year flood.

In Step 2, using a parametric description of the breaching, the dam is gradually removed from the topography, so that the water impounded behind the dam is released. The model computes the unsteady propagation of the flood
100 wave in the downstream valley."

Examples of results of Step 1 and Step 2 are displayed in Fig. R1 and Figs R2 to R5, respectively.

More details on the parametric description of the dam breaching are given in our reply to comment 2.3 below.

**Table R3.** Two-step hydraulic modelling protocol

|  | Hydraulic computation | Dam |
|---|---|---|
| Step 1 | Steady-state simulation | Incorporated in the DEM used for the simulation |
| Step 2 | Unsteady simulation | Gradually removed from the DEM (time-dependent topography) |

105

[Figure]

**Figure R1.** Longitudinal profile (in the dam area) of the bed and water levels for a steady discharge of 120 m³/s, as computed in Step 1 of the hydraulic modelling procedure ($k_s = 0.3$ m).

[Figure]

110    **Figure R2.** Longitudinal profiles of water levels computed in Step 2 of the hydraulic modelling procedure, assuming an instantaneous breaching of the dam (extreme case) and a flow rate of 120 m³/s in the river prior to dam breaching ($k_s = 0.3$ m).

[Figure]

**Figure R3.** Water depth distribution and velocity profiles before the breaching (a) as well as after 5 s (b), 10 s (c) and 20 s (d), as computed in Step 2 of the hydraulic modelling procedure. This computation assumes an instantaneous breaching of the dam (extreme case) and a flow rate of 120 m³/s in the river prior to dam breaching ($k_s = 0.3$ m).

[Figure]

**Figure R4.** Longitudinal profiles of water levels computed in Step 2 of the hydraulic modelling procedure, assuming a breaching duration of 600 s and a flow rate of 120 m³/s in the river prior to dam breaching ($k_s = 0.3$ m).

[Figure]

**Figure R5.** Longitudinal profiles of water levels computed in Step 2 of the hydraulic modelling procedure, assuming a breaching duration of 3600 s and a flow rate of 120 m³/s in the river prior to dam breaching ($k_s = 0.3$ m).

125  2.3. A highly critical issue is also the consideration of dam breach (lowering of the dam crest and release of the impounded water) – how does this work? Please explain! I have the feeling that you spend a lot of effort in describing base flow and lower boundary conditions at a high level of detail, but do not explain some of the really important aspects at all.

As stated in the initial manuscript, we closely followed the procedure proposed by Dewals et al. (2011) for
130  representing the dam breaching. Nonetheless, as pointed out by the Reviewer, we agree that this procedure deserves more explanations and more discussion in the manuscript, since it is indeed an important step of our study.

In our response below,

- we explicitly describe how the dam breaching is represented in the model;
- we also present and discuss the modelling results for an additional scenario of gradual dam breaching.

135  The text in the revised manuscript will be updated accordingly.

*"*The mechanisms of breaching of natural dams are complex, highly variable and incompletely understood. Hence, the modelling of the dam breaching may be a substantial source of uncertainty.

In the present study, process-oriented modelling of the breaching was not considered as a viable option, mainly due to the lack of detailed information on the dam material (graded, non-homogeneous material), the complexity of the
140  breaching of natural dams and the absence of validation data from similar case studies in the region. Instead, we opted for a simpler *parametric description* of the dam breaching which appears more consistent with the quality of available data and the overall level of uncertainty affecting the present study.

Among the various possible failure modes, we chose to represent dam *overtopping*, which is the most frequent failure mode for landslide dams. Failure induced by dam overtopping was reported for over 90 % of all landslide dams reviewed by Costa and Schuster (1988) and for 131 out of 144 cases reviewed by Peng and Zhang (2012).

As sketched in Fig. R6, the parametric breach model was implemented in the 2D flow model by means of a time varying topography. The breach outflow is thus explicitly computed by the flow model, enabling the representation of the hydraulic coupling between reservoir depletion, flow through the breach and possible backwater effects. This procedure requires a user-defined initial dam geometry (Fig. R6a) and a user-defined final geometry corresponding to the breached dam (Fig. R6e). In-between these two geometries, the algorithm performs a linear interpolation in time (Dewals et al. 2011). The breaching duration also needs to be prescribed by the user."

[Figure]

(a) Initial state (with dam)  (b) $t = 0.25\ T_f$  (c) $t = 0.50\ T_f$

(d) $t = 0.75\ T_f$  (e) Final state (without dam, $t = T_f$)

1000 m
992 m
984 m
976 m
968 m
960 m

25 m  0 m  100 m

**Figure R6.** Plane view of the topography evolution in the near-field of the landslide dam as a function of time ($T_f$ stands for the breach formation time).

Several prediction formulae have been tested for estimating the breaching duration (Froelich 2008, Peng and Zhang 2012, BREACH model …). They lead to scattered values, ranging in-between 10 min and one or two hours. Such discrepancies result from the limited number of real-world case studies for which information on breaching duration is available. For instance, out of a total of 1,239 cases reported by Peng and Zhang (2012), only 52 contain detailed information on the breaching and only 14 cases have records of breaching duration. Moreover, inconsistencies exist in these records, so that the regression results for breaching duration are generally less satisfactory (in terms of $R^2$) than for other breach parameters. These are the reasons why, in the revised manuscript, we will discuss the results obtained based on a range of plausible assumptions on the breaching duration: 10 min (Fig. R4) and 1 h (Fig. R5). One extreme assumption will also be considered (instantaneous dam failure) to characterize the envelope of possible results. The latter scenario could also correspond to an almost instantaneous breaching following an earthquake.

165 While the initial manuscript detailed only the results for the most extreme case (instantaneous dam failure), the revised version of the manuscript will include a detailed presentation of the results obtained for the other two breaching durations (10 min and 1 h). The text and all figures in section 3.3 will be revised accordingly. For instance, Figs. 11 to 13 and Tabs. 7 to 9 in the original manuscript will be replaced by the following figures and tables in the revised manuscript. The discussion will also be adapted, as the results reveal a substantial influence of
170 the breaching duration in the upper part of the valley; while this influence becomes much smaller in the urban area of interest.

[Figure]

**Figure 11.** Computed water depths (a, b) and discharge (c, d) for various pre-failure flow conditions (base flow, 20- and 50-year floods), and corresponding maximum water depths (a, b) and peak discharges (c, d) after dam breaching, in cross-sections 1 to 4 and for a roughness
175 height $k_s = 0.1$ m (a, c) and 0.3 m (b, d). 'Breach-induced flow_G10', 'Breach-induced flow_G60' and 'Breach-induced flow_I' stand for 'Breach-induced.flow_gradual with 10 minute as breaching time', 'Breach-induced.flow_gradual with 60 minute as breaching time' and 'Breach-induced.flow_instantaneous'.

[revised manuscript text omitted]

The text in the revised manuscript will be updated accordingly.

2.4. You claim to consider flood hazard – however, it is only the possible intensity which is used for the preparation of the maps – hazard would also have to include a measure for frequency. Some rewording (e.g. flood intensity indication map?) will be necessary.

We agree and we will revise the terminology throughout the manuscript, as it is already done in the legend of the maps within Fig. 13a and Fig. 13b proposed above.

2.5. You resample the 10x10 m DEM to 2x2 m. it is absolutely clear to me that this is necessary for numerical reasons – still, it does not increase the level of topographic detail. How wide is the river, i.e. is an effective 10x10 m cell size sufficient to capture the topographic patterns governing the flow? Please discuss.

In our response below, we first highlight that the study was conducted in a data-scarce context. Next, we report on a field survey conducted in the study area, which enables assessing the DEM we used for hydraulic modelling.

The average width of the river is about 20 m for a discharge of 3 m³/s, 32 m for 60 m³/s (20-year flood) and 40 m for 120 m³/s (50-year flood). Hence, a computational spacing of 2 m (obtained after resampling) is certainly fine enough to represent the flow field over the width of the river, since the number of computational cells over the width of the river is in-between 10 and 20. Nonetheless, the Reviewer is of course right that only the topographic details already present in the initial DEM (10 m × 10 m) are captured in the topography used for hydraulic modelling. This situation stems from the data-scarce environment in which this study was conducted, as also acknowledged by Reviewer 2.

In developed countries, light detection and ranging (LiDAR) elevation data are generally available at a high resolution (up to 0.5 m horizontally). In contrast, data for the study area are particularly scarce. Data scarcity is a common challenge in many regions in Africa. This reality was emphasized by various authors such as Jacobs et al. (2016) or Alvarez et al. (2017). Based on available elevation data (usually SRTM with a horizontal resolution of approximately 30 m), these authors performed hydraulic simulations leading to conclusions considered as scientifically relevant and recently published in leading international journals. This suggests that using medium- or low-resolution products remains a valuable intermediate step to advance our understanding of flood risk in data-scarce areas in Africa, provided that the results are interpreted in light of the uncertainties in input data. In this context, a 10 m resolution is among the best in the region, especially when compared to SRTM and ASTER GDEM provided by USGS. This is the reason why the 10 m × 10 m DEM was used in this manuscript (Section 2.2).

Besides, we conducted field surveys during the dry season (June-September) in 2014 and in 2015. The surveys covered the main riverbed and part of the floodplains (band of 10-20 m) of Kanyosha River, from 500 m upstream of the dam down to Lake Tanganyika. Fig. R7 shows the extent of the field survey, compared to the position of the banks of the river and to the limits of the 10 m × 10 m DEM used for hydraulic modelling. The available equipment did not allow measurements in the lake (this is the reason why we present in the manuscript a sensitivity analysis with respect to the downstream boundary condition).

As shown in Fig. R8, the differences between the DEM used in our hydraulic simulations and data from the field survey remain moderate, as they range generally between – 0.5 m and + 0.5 m. The median and mean differences are both - 7 cm. The RMS error between DTM 10 m × 10 m and field measurements is 65 cm and seems reasonable.

Most significant differences are obtained near the river banks, which may result from discretization errors and/or from the instability of the banks due to planform evolution of the riverbed over the period from 2012 (when the 10 m × 10 m DEM was produced) to 2014 (field survey in the main riverbed).

[Figure]

320 **Figure R7.** Extent of the field measurements ( ), of the river banks (—) and of the computational domain: (a) lower part of the valley, (b) middle part and (c) upper part.

[Figure]

**Figure R8.** Elevation difference between the topography from field measurement and the resampled 2 m × 2 m DEM used for hydraulic modelling.

325 We also expect that in the upper part of the valley, which shows a distinctive V-shape with relatively steep lateral slopes, as the flow tends to concentrate in the main canal and its vicinity, the hydraulic modelling results are less affected by small inaccuracies in the DEM than further downstream.

In the revised version of the manuscript, we will:

- discuss the resolution of the original and resampled DEMs with respect to the river width;
330 - highlight that this study was conducted in a "data-scarce" context and discuss the implications in terms of reliability of the results in both the upper and lower parts of the valley;
- refer to the field survey to appreciate the reliability of the topographic data;
- explicitly state that using higher resolution and updated elevation data (particularly for the river bathymetry) is a necessary next step of this research.

335 These important points will be added in Section 2.2 (topographic and geophysical data) and in section 4.3. of the revised version of the manuscript. We also propose to report in the revised manuscript on additional simulations performed based on the surveyed topographic data instead of the original 10 m × 10 m DEM for appreciating the sensitivity of the simulation results (peak discharge, inundation extent, water depths) to the inaccuracy in the topographic data.

340 2.6. The discussion on the uncertainties involved is very short, given that the uncertainty issue is very important when it comes to computer simulations. Some further considerations would be desirable (geotechnical parameters, hydrograph, sediment transport, ...).

In the revised manuscript, we will substantially expand our discussion on the model uncertainties (section 4.3 in the original manuscript), so that it becomes more representative of the whole spectrum of sources of uncertainties.
345 To make the discussion more structured, we will categorize the various uncertainties affecting our results as a function of their cause: *(i)* input data, *(ii)* model structure (i.e. processes which are incompletely represented in the model), *(iii)* model parameters and *(iv)* scenarios. Among others, we will refer to the aspects detailed hereafter.

- The influence of the *topographic and bathymetric data* will be discussed, in line with our response to comment 2.5 above.

350 - Another major and specific local challenge relates to the *planform variations of the river channel*. The banks of the Kanyosha River, like those of other rivers in Bujumbura, are not stabilized and frequently undergo strong changes due to erosion and anthropogenic disturbances. This results in changes of the river cross section and may affect the flow dynamics.

- The influence of sediment transport and morphodynamics will be discussed in line with our response to
355 comment 2.1.

- Moreover, the characteristic size of the bottom irregularities was observed to vary along the river channel. Therefore, although we tested different values of the friction coefficient in our simulations, uncertainties remain regarding the effect of the *spatial variability in bottom roughness*.

- In our simulations, we assume that the reservoir behind the dam is completely filled when the failure starts.
360 The actual situation could be different, as the breaching may occur before the complete filling of the reservoir. However, in such a case, the severity of the induced flooding would be lower, so that our assumption makes sense from the perspective of risk management. Filling of the reservoir takes about

5.5 hours, 17 minutes and 9 minutes in, respectively, the base flow scenario, the 20-year flood scenario, and the 50-year flood scenario.

365     •   In addition, the *dam breaching mechanism and dynamics* depends on a series of factors related to the resistance of the natural dam. The detailed prediction of this resistance is out of the scope of the present study (in which we *assume* a breach formation time); but it may considerably affect the actual breaching and the induced flood wave. Therefore, in the revised manuscript, we will detail three failure scenarios:

370     −   relatively slow gradual failure of the dam (60 min), initiated by the flow overtopping the dam after filling of the reservoir;

    −   relatively fast gradual failure of the dam (10 min);

    −   instantaneous failure (extreme scenario), resulting for example from the occurrence of a major disturbance like an earthquake.

Intermediate scenarios may also be considered if deemed relevant.

375
* * *
2.7. Flow depth x velocity does not result in m/s$^2$, but in m$^2$/s.
* * *
Indeed, this was a mistake and it will be corrected in the revised version of the manuscript ("m/s$^2$" will be replaced by "m$^2$/s" at Line 200 and Line 202 of the manuscript).

---

## Author Response (AR1)

**COVER LETTER**

Dear Editor,

Thank you very much for handling our manuscript: "Formation, breaching and flood consequences of a landslide dam near Bujumbura, Burundi". We deeply appreciate the Reviewers comments and suggestions, which have enabled us to substantially improve the quality of the manuscript.

In particular, we have introduced the following main revisions:

- extra simulations based on an alternate DEM (derived from a recent field survey) have been undertaken and compared with results of simulations using the initial DEM;
- additional breaching scenarios are now considered, involving various breaching times;
- a discussion on the influence of the amount of released solid material has been elaborated;
- the flow modelling procedure and the breach modelling are now more thoroughly described, with a number of additional figures provided as Supplements;
- material initially presented in Appendix, together with extra material, have been moved to Supplements to avoid an excessive length of the manuscript.

In the following, we present a point-by-point response to all the comments raised by the Reviewers. To a great extent, the present document consists in an updated version of our responses provided during the discussion phase, in which we also describe how the revisions have indeed been incorporated in the manuscript. Additional simulations, which were planed during the discussion phase, have now been performed and, in the revised manuscript, we detail the findings obtained from these new computations.

Best regards,

L. Nibigira, Corresponding author

**Reviewer 1 (M. Mergili)**

Corresponding changes are highlighted in green in the revised manuscript.

**General comment**

The authors present an interesting study on the possible impact of the dynamics of a landslide on flooding downstream, thereby considering the effects of breaching of a landslide dam. Much of the manuscript is well written, structured, and illustrated. Particularly the evaluation of the stability conditions of the landslide is very well described. However, coupling of landslide and flood is, in my opinion, insufficiently covered. Particularly in this context I have identified some important issues requiring improvements and therefore recommend major revisions. I now outline my suggestions and comments in the order of decreasing priority:

Thank you for the attention paid to our manuscript.

**Specific comments**

2.1. One of my major concerns relates to the fact that only flooding by water is considered. Breach of the landslide dam would release a huge amount of solid material (most probably deeply weathered tropical soil) that would be incorporated in the flow and could possibly lead to completely different characteristics and downstream impact of flooding, compared to clear water flow. This issue is not even discussed at all. I see two possibilities to face this challenge: (i) incorporating sediment load in the flow simulation; or (ii) a thorough argumentation and discussion why this is not necessary. Either (i) or (ii) should be an absolute requirement for the acceptance of the manuscript.

We indeed only included water in the flood wave computation, while breaching of the landslide will release a substantial amount of solid material. The actual flow will have an intermediate behavior between clear-water flow and debris or granular flow. In the revised manuscript, we address this issue as follows:

- this assumption is now acknowledged explicitly in the revised manuscript (beginning of section 2.4);
- the reasons for this (data-scarce context) and its implications are discussed as detailed hereafter;
- we also propose a sensitivity analysis to appreciate some of the consequences of this assumption;
- we clearly state in the Conclusion that including sediment transport in the simulation is a valuable direction for future work in this region.

The following discussion has been included at the beginning of Section 2.4 of the revised manuscript:

"We only included water in the flood wave computation, while the actual breaching of the landslide dam would release a substantial amount of solid material. The real flow would have an intermediate behaviour between clear-water flow and debris or granular flow. As shown in Table S2 (Supplement 3), some recent studies neglected sediment transport in the analysis of floods induced by the breaching of landslide dams (Fan et al., 2012; Yang et al., 2013), while others did take sediment transport into account (Li et al., 2011; Shrestha and Nakagawa, 2016) since it may have considerable implications on the volume of mobilized material as well as on morphological

evolutions of the valley bottom (e.g., sediment deposition). Nonetheless, we believe that, in the context of the present study, going for more complexity in the modelling framework (i.e. including sediment transport) would mainly produce more speculative results because validation data are neither available for our case study nor for any similar one in the region, which remains largely understudied. Table S2 shows that previous studies which considered sediment transport benefited all from available validation data, such as observed flood discharges or depths of sediment deposits."

In addition, this discussion is further expanded in Subsection 4.3.4 of the revised manuscript. In particular, we take benefit there of an additional set of simulations undertaken to test the sensitivity of the modelling results to the use of a different DEM (derived from field survey, see Subsection 4.3.2). These additional simulations give some insights into the plausible effect of changes in the river bathymetry (e.g. as could be obtained as a result of erosion or deposition, which is not modelled explicitly here) and we suggest to link it to the effect of sediment transport.

Finally, we clearly state in the revised Conclusion that the present study should be further continued using more advanced debris flow / granular flow modelling tools such as presented by Mergili et al. (2012a, 2012b, 2017) or others, and adapted to channelized debris flow.
* * *
> 2.2. I do not fully understand the work flow of the flood modelling: in the first step, do you (i) simulate the base flow without the dam incorporated, or do you (ii) fill the lake behind the dam to let it flow out in the second step? The description in Sect. 2.4.3 is confusing and has to be improved.

In the first step, we fill the lake behind the dam to let it flow in the second step. This is now clarified in the revised manuscript, by introducing a new table (Table 3) and by rewording section 2.4.3 as detailed hereafter:

"The hydraulic simulations aim at evaluating the impact of the dam failure as a result of the water impoundment behind it and the river overflowing the dam crest. Thus, the initial step of hydraulic modeling considers a filled reservoir and a steady flow of water over the crest of the dam before failure. In line with Dewals et al. (2011), the modelling procedure involves two steps:

- step 1: a pre-failure steady flow is computed in the river, under three different hydrological scenarios (steady flow corresponding to the mean discharge in the river or to a 20-year flood, or a 50-year flood);
- step 2: using the result of step 1 as initial condition, the flow induced by the breaching of the dam is computed.

In Step 1, the dam geometry is incorporated in the topographic data used for flow computation. This means that the dynamics of material sliding into the river is not explicitly reproduced in the hydraulic modelling. As it is not possible to anticipate when the landslide dam breaching might occur, we consider three different pre-failure flow conditions: base flow, 20-year flood and 50-year flood.

In Step 2, using a parametric description of the breaching, the dam is gradually removed from the topography, so that the water impounded behind the dam is released. The model computes the unsteady propagation of the flood wave in the downstream valley."

Examples of results of Step 1 and Step 2 are displayed in Fig. S2 and Figs S3 to S6, in Supplement 6.

95    More details on the parametric description of the dam breaching are given in our reply to comment 2.3 below.

2.3. A highly critical issue is also the consideration of dam breach (lowering of the dam crest and release of the impounded water) – how does this work? Please explain! I have the feeling that you spend a lot of effort in describing base flow and lower boundary conditions at a high level of detail, but do not explain some of the really important aspects at all.

100    As stated in the initial manuscript, we closely followed the procedure proposed by Dewals et al. (2011) for representing the dam breaching. Nonetheless, as pointed out by the Reviewer, we agree that this procedure deserves more explanations and more discussion in the manuscript, since it is indeed a key step of our study.

In our response below,
- we explicitly describe how the dam breaching is represented in the model;
105    - we also present and discuss the modelling results for an additional scenarios of gradual dam breaching.

The text in the revised manuscript has been updated as follows (end of Section 2.4.3, and new Fig. 9):

*"The mechanisms of breaching of natural dams are complex, highly variable and incompletely understood. Hence, the modelling of the dam breaching may be a substantial source of uncertainty. In the present study, process-oriented modelling of the breaching was not considered as a viable option, mainly due to the lack of detailed
110    information on the dam material (graded, non-homogeneous material), the complexity of the breaching of natural dams and the absence of validation data from similar case studies in the region. Instead, we opted for a simpler *parametric description* of the dam breaching which appears more consistent with the quality of available data and the overall level of uncertainty affecting the present study.

Among the various possible failure modes, we chose to represent dam *overtopping*, which is the most frequent
115    failure mode for landslide dams. Failure induced by dam overtopping was reported for over 90 % of all landslide dams reviewed by Costa and Schuster (1988) and for 131 out of 144 cases reviewed by Peng and Zhang (2012).

As sketched in Fig. 9, the parametric breach model was implemented in the 2D flow model by means of a time varying topography. The breach outflow is thus explicitly computed by the flow model, enabling the representation of the hydraulic coupling between reservoir depletion, flow through the breach and possible backwater effects. This
120    procedure requires a user-defined initial dam geometry (Fig. 9a) and a user-defined final geometry corresponding to the breached dam (Fig. 9e). In-between these two geometries, the algorithm performs a linear interpolation in time (Dewals et al. 2011). The breaching duration also needs to be prescribed by the user.

Several prediction formulae have been tested for estimating the breaching duration (Froehlich 2008, Peng and Zhang 2012, BREACH model …). They lead to scattered values, ranging in-between 10 min and one or two hours.
125    Such discrepancies result from the limited number of real-world case studies for which information on breaching duration is available. For instance, out of a total of 1,239 cases reported by Peng and Zhang (2012), only 52 contain detailed information on the breaching and only 14 cases have records of breaching duration. Moreover, inconsistencies exist in these records, so that the regression results for breaching duration are generally less satisfactory (in terms of $R^2$) than for other breach parameters. These are the reasons why, in the revised manuscript,
130    we will discuss the results obtained based on a range of plausible assumptions on the breaching duration, in-between 10 min and 1 h. One extreme assumption was also tested (instantaneous dam failure) to characterize the envelope

of possible results. The latter scenario could also correspond to an almost instantaneous breaching as a result of an earthquake."

While the initial manuscript detailed only the results for the most extreme case, the revised version of the manuscript includes now also a detailed presentation of the results obtained for the other two breaching durations. The text and all figures in Section 3.3 have been revised accordingly. This includes Figs. 13 to 15 and Tabs. 8 to 10 in the revised manuscript, which have been updated compared to their initial version (Figs. 11 to 13 and Tabs. 7 to 9 in the original manuscript). The main text and discussion have also been adapted (in Sections 3.3 and 3.4), as the results reveal a substantial influence of the breaching duration in the upper part of the valley; while this influence becomes much smaller in the urban area of interest. Multiple other adjustments have also been made.
* * *
2.4. You claim to consider flood hazard – however, it is only the possible intensity which is used for the preparation of the maps – hazard would also have to include a measure for frequency. Some rewording (e.g. flood intensity indication map?) will be necessary.
* * *
We agree. Throughout the manuscript, the terminology "hazard" has been replaced by "flood intensity" (e.g., legends of maps in Figs 15a and b).
* * *
2.5. You resample the 10x10 m DEM to 2x2 m. it is absolutely clear to me that this is necessary for numerical reasons – still, it does not increase the level of topographic detail. How wide is the river, i.e. is an effective 10x10 m cell size sufficient to capture the topographic patterns governing the flow? Please discuss.
* * *
In our response below, we first highlight that the study was conducted in a data-scarce context. Next, we report on a field survey conducted in the study area, which enables assessing the DEM we used for hydraulic modelling. Corresponding changes are included in Section 2.2 of the revised manuscript.

- The average width of the river is about 20 m for a discharge of 3 m³/s, 32 m for 60 m³/s (20-year flood) and 40 m for 120 m³/s (50-year flood). Hence, a computational spacing of 2 m (obtained after resampling) is certainly fine enough to represent the flow field over the width of the river, since the number of computational cells over the width of the river is in-between 10 and 20. Nonetheless, the Reviewer is of course right that only the topographic details already present in the initial DEM (10 m × 10 m) are captured in the topography used for hydraulic modelling. This situation stems from the data-scarce environment in which this study was conducted, as also acknowledged by Reviewer 2.
- In developed countries, light detection and ranging (LiDAR) elevation data are generally available at a high resolution (up to 0.5 m horizontally). In contrast, data for the study area are particularly scarce. Data scarcity is a common challenge in many regions in Africa. This reality was emphasized by various authors such as Jacobs et al. (2016) or Alvarez et al. (2017). Based on available elevation data (usually SRTM with a horizontal resolution of approximately 30 m), these authors performed hydraulic simulations leading to conclusions considered as scientifically relevant and recently published in leading international journals. This suggests that using medium- or low-resolution products remains a valuable intermediate step to advance our understanding of flood risk in data-scarce areas in Africa, provided that the results are

interpreted in light of the uncertainties in input data. In this context, a 10 m resolution is among the best in the region, especially when compared to SRTM and ASTER GDEM provided by USGS. This is the reason why the 10 m × 10 m DEM was used in this manuscript (Section 2.2).

- Besides, we conducted field surveys during the dry season (June-September) in 2014 and in 2015. As shown in Fig. R1, the surveys covered the main riverbed and part of the floodplains (band of 10-20 m) of Kanyosha River, from 500 m upstream of the dam down to Lake Tanganyika. Fig. R1 shows the extent of the field survey, compared to the position of the banks of the river and to the limits of the 10 m × 10 m DEM used for hydraulic modelling. The available equipment did not allow measurements in the lake (this is the reason why we present in the manuscript a sensitivity analysis with respect to the downstream boundary condition).

- As shown in Fig. 4 (in the revised manuscript), the differences between the DEM used in our hydraulic simulations and data from the field survey remain moderate, as they range generally between – 0.5 m and + 0.5 m. The median and mean differences are both - 7 cm. The RMS error between DTM 10 m × 10 m and field measurements is 65 cm, which seems reasonable. Most significant differences are obtained near the river banks, which may result from discretization errors and/or from the instability of the banks due to planform evolution of the riverbed over the period from 2012 (when the 10 m × 10 m DEM was produced) to 2014 (field survey in the main riverbed).

[Figure]

**Figure R1.** Extent of the field measurements (  ), of the river banks (—) and of the computational domain: (a) lower part of the valley, (b) middle part and (c) upper part.

We also expect that in the upper part of the valley, which shows a distinctive V-shape with relatively steep lateral slopes, as the flow tends to concentrate in the main canal and its vicinity, the hydraulic modelling results are less affected by small inaccuracies in the DEM than further downstream.

In brief, in Section 2.2 of the revised manuscript, we:

- discuss the resolution of the original and resampled DEMs with respect to the river width;
- highlight that this study was conducted in a "data-scarce" context and discuss the implications in terms of reliability of the results in both the upper and lower parts of the valley;
- refer to the field survey to appreciate the reliability of the topographic data;
- explicitly state that using higher resolution and updated elevation data (particularly for the river bathymetry) is a necessary next step of this research.

In sections 4.3.2 of the revised manuscript (as well as in the new Supplement 7), we present and discuss the results of additional simulations performed based on the surveyed topographic data instead of the original 10 m × 10 m DEM. This enables appreciating the sensitivity of the simulation results (peak discharge, inundation extent, water depths) to variations in the topographic data.

205 Finally, the need to devote more efforts to the collection of more accurate topographic data in the case study area is stressed in the Conclusion.

> 2.6. The discussion on the uncertainties involved is very short, given that the uncertainty issue is very important when it comes to computer simulations. Some further considerations would be desirable (geotechnical parameters, hydrograph, sediment transport, ...).

210

In the revised manuscript, we have substantially expanded our discussion on the model uncertainties (Section 4.3), so that it becomes more representative of the whole spectrum of sources of uncertainties. Among others, we refer to the aspects detailed hereafter.

- In our simulations, we assumed that the reservoir behind the dam is completely filled when the failure
215 starts. The actual situation could be different, as the breaching may occur before the complete filling of the reservoir. However, in such a case, the severity of the induced flooding would be lower, so that our assumption makes sense from the perspective of risk management. Filling of the reservoir takes about 5.5 hours, 17 minutes and 9 minutes in, respectively, the base flow scenario, the 20-year flood scenario, and the 50-year flood scenario. This is now detailed in Subsection 4.3.1.

220 - Moreover, the characteristic size of the bottom irregularities was observed to vary along the river channel. Therefore, although we tested different values of the friction coefficient in our simulations, uncertainties remain regarding the effect of the *spatial variability in bottom roughness* (Subsection 4.3.1).

- Another major and specific local challenge relates to the *planform variations of the river channel*. The banks of the Kanyosha River, like those of other rivers in Bujumbura, are not stabilized and frequently
225 undergo strong changes due to erosion and anthropogenic disturbances. This results in changes of the river cross-section and may affect the flow dynamics (also referred to in Subsection 4.3.2).

- The influence of the *topographic and bathymetric data* are now discussed in Subsections 4.3.2 and 4.3.3, in line with our response to comment 2.5 above.

- The influence of sediment transport and morphodynamics is now discussed in Subsection 4.3.3, in line
230 with our response to comment 2.1.

- In addition, the *dam breaching mechanism and dynamics* depends on a series of factors related to the resistance of the natural dam. The detailed prediction of this resistance is out of the scope of the present study (in which we *assume* a breach formation time); but it may considerably affect the actual breaching and the induced flood wave. Therefore, in the revised manuscript, we detailed three failure scenarios:

235 – relatively slow gradual failure of the dam (60 min), initiated by the flow overtopping the dam after filling of the reservoir;

– relatively fast gradual failure of the dam (10 min);

– instantaneous failure (extreme scenario), resulting for example from the occurrence of a major disturbance like an earthquake.

240

| 2.7. Flow depth x velocity does not result in m/s$^2$, but in m$^2$/s. |
|---|

Indeed, this was a mistake and it has now been corrected throughout the revised version of the manuscript: "m/s$^2$" has been replaced by "m$^2$ s$^{-1}$".

**Reviewer 2 (O. Dewitte)**

Corresponding changes are highlighted in yellow in the revised manuscript.

**General comment**

Dear colleagues,

The research proposed by Nibigira and co-authors focuses on landslide damming and possible subsequent flood occurrence in Bujumbura. More specifically, the authors focus on a large landslide that develops in a watershed flowing toward the city center. The first part deals with geophysics and landslide 3D and 2 D reconstruction. The second part is with the modelling of the landslide. The third part deals with the simulations of the floods and their associated hazard induced by dam breaching. The manuscript presents interesting results for such an understudied region; especially when one considers that in these African regions data scarcity is commonplace. I agree with the comments and suggestions by the first reviewer. I have therefore done my review accordingly (see supplement material). This includes minor to moderate comments and technical corrections. Two points I insist to are the weathering of the lithology and, for the introduction, the data-scare context and methodological challenges of the study area.

Regards

Olivier Dewitte

Thank you for these interesting and inspiring comments and suggestions on our manuscript.

**Specific comments**

2. 1: (Line 20) The introduction can be improved as it suffers from a lack of state of the art referencing to the general literature on landslide dam and related flood modelling. As it stands now we miss the broader context of this study (scientific and societal needs to perform such a study, challenges to get relevant input data for modelling, context of data scarcity in the studied region, methodological challenges, etc.). Then, based on this context, comes the objectives that you aim at studying a specific landslide in a specific region.

The Introduction has been entirely reformulated according to the Reviewer's comments.

275

> 2. 2: (Fig. 1) The relief is represented with shiny colors. I would have combined them with a hill shade view to improve readability.

To take into account this very relevant recommendations, the map has been modified (see Fig. 1a in the revised manuscript).

> 2. 3: (Fig. 1) Delete the "_"

280 The "_" is deleted from the legend of Fig. 1a.

> 2.4: (Fig. 1) Color issues: blue for watershed delineation and red for a river is weird. I suggest blue for the river and another color for the watershed.

These suggestions have been taken into account: blue is used now for the river network while red is used for the
285 watershed delineation (Fig. 1a).

> 2.5: (Fig. 2) where is the waterfall located?

This waterfall formed on a former flood control structure is located at 270 m downstream of cross-section 3 shown in Fig. 3 of the revised manuscript. This is now stated in caption of Fig. 1.

290

> 2.6: (Line 50) This means that the dotted contour line used in Figure 2 (Fig. 1b in the revised manuscript) represents only part of the landslide; i.e. the landslide section that you consider in the modelling. That something that should be stressed better within the Figure caption. So far, this figure shows some landslide parts such as the one where the compass rose is positioned that are not delineated alone. Based on the figure and its caption alone, it looks
295 strange.

This is indeed very relevant, since other instabilities also appear in the figure. The whole site of the BTL is strongly weakened, creating small secondary landslides juxtaposed with the main landslide. These are located at the foot of the slope along the Kanyosha River and do not follow the dynamics of the global movement. They are detached by small blocks that are then subject to erosion and are therefore not likely to produce significant effects on the risk
300 of flooding. This is well expressed from line 48 to line 51 of the original manuscript.

In our opinion, the confusion comes partly from the fact that the figure was not adequately introduced in the main text. For this reason, we have removed the paragraph from Line 48 to Line 51 of the original manuscript ("Our recent observations ... move and form a landslide dam") and we have inserted an improved version of this paragraph just before Fig. 2 in the revised manuscript. This helps the reader to better understand Fig. 2 (Fig. 1b in the revised

305    manuscript). In the revised manuscript, we have also added more information in the caption of Fig. 1 caption. The instabilities are now delimited (blue and yellow contours) in Fig. 1b (figure caption was updated accordingly).

> 2.7: (Line 51) The presence of water ponds is a sign of the landslide current activity. However, can you elaborate more on the role of those ponds on future instability? Do you have references to support this?

First, this idea is strongly supported later by our results. In fact, these ponds contribute to the saturation of the
310    landslide body, whereas in sections 3.1 and 3.2, we have quantified the impact of groundwater on landslide, as it greatly reduces the factor of safety (Table 5) and increases X-acceleration.

The weight of water ponds also contributes to the loading of the sliding. The impact of the slope overloading on the landslide dynamics is also strongly highlighted both by Terzaghi (1950), Varnes (1978), Popescu (1994) and Popescu (2002).

315

> 2.8: (Line 53) well, I wonder how one single landslide could have on impact in the country's economy. I suggest to rephrase this in a more balanced way: "... makes it a potential danger for people and infrastructures of that area".

Line 53 of the original manuscript has been reworded as follows: "This landslide was chosen for its size (it is the largest active landslide in the vicinity of Bujumbura with a volume of more than $4 \times 10^6$ m$^3$) and due to its position
320    along the Kanyosha River, upstream of the city (Fig. 1) making it a potential danger for people and infrastructures in the area".

Our initial point was inspired from the disastrous consequences of the landslide along the Gasenyi River in February 2014. There was a real disruption at least for the short term. More than 940 homes were destroyed and 12,500 people were homeless. Reconstruction and studies as well as corollary changes to operate to avoid new disasters in
325    the area etc., have required a great effort in the context of a fragile economy. The BTL is more than three times larger than that of the Gasenyi River (under 10 m high for the Gasenyi landslide dam, while the BTL can potentially be 15 m-20 m high).

The following information gives some insights into the cost to the economy of the flood of 2014:

- for 2014, the Government of Burundi adopted an austerity budget policy (called "*Gutubika Uwavubi*"),
330      with an annual budget of 1403.3 billion Burundian Francs (909.8 million USD);
- following the catastrophic floods in February 2014, a joint mission of the United Nations, the European Union, the African Development Bank and the World Bank was deployed to evaluate the disaster;
- the losses related to public infrastructure, crops and private houses were estimated at 18.9+ Million USD, which is about 2.1% (of which 0.49% for the public infrastructures and 1.6% for crops and private houses)
335      of the annual budget of the State.

However, the rehabilitation and management measures and especially the resulting disaster prevention measures are much cheaper than the initial damage. These amounts were estimated at 105.7 million USD, equivalent to 11.61 % of the annual budget. This is a tremendous challenge in the economic context mentioned above.

2.9: (Line 62: channel description) I know that they are some definitions related to the size. I am not sure a 40 cm diameter is appropriate for a pebble. Not even 10 cm to be checked with a classification system adapted for your study.

In the revised manuscript, the extended Udden-Wentworth grain-size scale nomenclature has been applied, as proposed also in our reply to comment 2.12. The channel description (Sec. 2.1) has been reworded accordingly.

2.10: (Line 63: channel description) size of those stones? Usually, above few tens of cm we call them boulders.

Section 2.1 has been reworded, consistently with our reply to Comment 2.12.

2.11: (Line 65: channel description) Accumulation zones of what?

This section has been improved by the use of a suitable grain-size classification. The accumulation consists of fine materials (clay and silt) trapped in small flats or generally upstream of the remains of old hydraulic structures. Often, within these areas, small green islets developed like those shown in Fig. 1c in the revised manuscript. The rewording proposed in our reply to the comment 2.12 includes all these details.

2.12: (Line 65: channel description) Do you make a distinction between pebbles and debris? If so, where does debris come from? More info needed.

Regarding the comments from 2.7 to 2.10 and their reply given above, a rewording of the channel description (Sec. 2.1) is proposed below:

*"The Kanyosha River main channel has deposits with variable grain size. Based on the extended Udden-Wentworth grain-size scale nomenclature (Terry and Goff, 2013), the riverbed material can be classified into three main groups. The first consists of cobbles of around 10 cm in diameter or more (Fig. 2c). The coarse part of this category consists of fine boulders, with a diameter generally under 40 cm. The second group is made up of isolated medium boulders that are often prone to the action of humans, carving them into building materials (mainly paving plates).*

*This category is difficult to take into account due to its strong irregularity. The third group consists of silt and clay zones, generally near former hydraulic structures in the downstream part of the river. In this category, we can mention small herbaceous islets, often located near the river overbanks. As in the second group, this category is found only in small isolated and scattered areas, subject to strong seasonal variations. Globally, the first group remains hydraulically predominant. Here, the variability of the grain size was accounted for by means of sensitivity analysis (Sec. 3.3).*

*In 2006, hydraulic structures were constructed to regulate the river; but they were quickly damaged by floods during the following raining seasons. Nonetheless, isolated cobbles resulting from the destruction of these structures are observed. They join the second group described above. The accumulation of material upstream of the remains of the structures often form horizontal platforms, generating small waterfalls (Fig. 2b)."*

Details for the Udden-Wentworth grain-size scale nomenclature are provided in Supplement 1 (Table S1).

> 2.13: (Fig. 3) Colors are not used at their best. Suggestions for improvement: for elevation: avoid blue. As you are studying flood, it is pretty confusing. If you do not use blue for topography, you can use it for the river (which sounds more logical). Then the cross-sections can be in black.
>
> 2.14: (Fig. 3_ caption) As it reads now, it is pretty confusing. It seems that you have used a hillshade product for the modelling. In addition, the figure shows a mix of DEM and hillshade.
>
> 2.15: (Fig. 3_caption) No need to repeat in the caption what you have in the legend.

The requested changes have been incorporated in the new Fig. 3 and its caption in the revised manuscript.

> 2.16: (Line 94) Up to now, there is no reference in the landslide description that the material is made of rock. From figure 2, it looks as if most of the landslide material is highly weathered. I think it is important to mention this weathering issue as this is something very specific of this type of tropical climate.

The original material of BTL is a gneiss which, by the alteration, is partially transformed into a clay on the surface. The depth of the altered layer is about 20 m.

The study area experiences alternations between dry and rainy seasons. The long dry season (from June to September) is followed by the small rainy season (from October to December), then by the small dry season (during the months of January and February). The cycle ends with the strong rainy season from March to May, just before the return of the dry season. Since the photos in Fig. 2 were taken in October, the ground was relatively wet, but not as much as in December and during the strong rainy season.

Especially for the low parts of the landslide, the humidity is never very low due to the recharge of the water table by the ponds of water located on the landslide. On the other hand, the groundwater recharge follows the dynamics of the seasons. In the context mentioned above, the action of the rainy season in the body of the landslide is quickly sensible, due to the water that sneaks in the interstices.

This additional information is included in the revised Section 2.3.

> 2.17: (Line 96) GPS? Why not deriving the profile from the DEM?

The geophysical survey technique by electric tomography profiles requires elevation values at each of the electrodes positioned every 5 m. Thus, additional elevation values were needed to complete the $10 \times 10$ m DEM.

2.18: (Line 110) Can you say something about the material and the weathering conditions?

Indeed, the details given in response to comment 2.16 above in terms of material and weathering conditions have been inserted in Section 2.3 of the revised manuscript, as follows:

405

410

"*The original material of BTL is a gneiss which, by the alteration, is partially transformed into a clay on the surface. The depth of the altered layer is about 20 m. The study area experiences alternations between dry and rainy seasons. The long dry season (from June to September) is followed by the small rainy season (from October to December), then by the small dry season (during the months of January and February). The cycle ends with the strong rainy season from March to May, just before the return of the dry season. Since the photos in Fig. 2 were taken in October, the ground was relatively wet, but not quite enough compared to December and the strong rainy season. Especially for the lower parts of the landslide, the humidity is never very low due to the recharge of the water table by the ponds of water located on the landslide. On the other hand, the groundwater recharge follows the dynamics of the seasons. In the context mentioned above, the action of the rainy season in the body of the landslide is quickly sensible, due to the water that sneaks in the interstices*.".

415

2.19: (Fig. 6) Not visible (The point 14)

2.20: (Fig. 6) X scale of Figure 5 starts with 0 close to the landslide main scarp and the river axis is at 900 m… I suggest the same scale is used here for the sake of clarity.

The requested changes in Fig. 6 of the original manuscript (Fig. 7 in the revised manuscript) have been incorporated: *x* scale corrected and point 14 visible.

420

2.21: (Line 140) References that justify these options? Can you discuss this?

Even tough each case is different, data availability helps to refer to an existing data base and case studies within the study area. Unfortunately, considering the context of data scarcity in the region, it is not easy to find related references. This is why we did not assign a single value, but 4 different shaking duration values to well illustrate
425 the behavior of the model corresponding to different scenarios. The seismic context is analyzed on the basis of earthquakes data from the Global Seismographic Network stations of the Incorporated Research Institutions for Seismology (IRIS) on the Lake Tanganyika Region. Therefore, based on that situation and local site effects, the 0.1g-wavelet used was chosen as a reasonable value to predict the behavior of the landslide under very moderate seismic shaking.

430 Corresponding changes have been included at the end of Section 2.3 of the revised manuscript.

2.22: (Line 264) Replace by "could".

This has been corrected in the revised version of the manuscript.

435

2.23: (Line 265) The origin of the water pounds present in the landslide can also be of runoff concentration origin.

Yes, we do agree. Actually, this was already mentioned in the discussion section of the initial version of the manuscript (Line 399): "Due to the landslide surface morphology, water could accumulate at its surface and form some ponds (see view of main pond in Fig. 2a)."

440

2.24: (Line 265) You mean, in the landslide body?

We mean water ponds on the landslide (BTL), as shown in Fig. 2a in the revised manuscript.

2.25: (Line 269) You mean BTL landslide is it?

Yes, indeed. To avoid any confusion, we have changes "Kanyosha Landslide" into "BTL".

445

2.26: (Line 271) Have you observed/measured this, or are they hypothesis. For such a large landslide, the response of the displaced material can be more complex that just following the seasonality of the wet periods.

This statement is not limited to the BTL, but it applies to most landslides in Bujumbura, especially those located along the Mugere River (in the south of the City Bujumbura) and Muha River (north of the Kanyosha River). From 450 2012 to 2016, field investigations were carried out to assess the risk of landslides. Benchmarks had been installed at some specific scarps and cracks' boundaries. Their relative position change and distances were measured over time to follow the movement of the ground. Therefore, this is based on field observations and measurements.

2.27: (Line 397) Nowhere earlier clay is mentioned as being a component of the landslide material. Some info 455 needs to be provided earlier.

In the revised manuscript, clay layer is mentioned in Fig. 2c and its caption, as well as in Sections 2.1 and 2.3.

> 2.28: (Fig. 14) This figure must appear much earlier in the text, when description of the landslide is made. This figure does not provide results s.s. and in addition is called just after Figure 2.

This Figure has now been inserted in the Introduction (Section 1), as Fig. 2.

> 2.29: (Fig. 14_Caption) Is this important to focus on the river bed material in this context?

Indeed, it is not important here, as the main information of relevance is the orientation of layers. Therefore, in the revised version, we do not repeat information on the bed grain-size, which is already given by Fig. 1d and described extensively in Section 2.1.

> 2.30: (Line 436) To add a comment to those of reviewer 1, discussion should also focus on the extension of the hazard zone with regard to the use of a 10 m resolution DEM. In addition, being familiar with the region, I know that river banks can sometimes be of several meters high, i.e. well above the water depth that you modelled. Can you comment on that as well?

We do agree that the river banks can sometimes be several meters deeper.

Nonetheless, Figure 13a and b of the revised manuscript show that, for scenarios leading to flooding outside the main channel (e.g. breach-induced flow), the water depth values reach 3.5 m - 4 m in the upstream parts (cross-sections 1 and 2) and 1.8 m – 2 m downstream (cross-sections 3 and 4). Particularly, in the urban area, the main channel depth is relatively shallow and, at some locations, a water depth exceeding 1.5 m is sufficient to lead to a considerable flood extent in the floodplains.

Moreover, the water depth values displayed at the cross section do not include the maximum values over the entire simulation domain (which reach up to 15.95 m). This is visible in Fig. R2: the cross-sections do not correspond to the location of maximum computed water depths.

The issue raised here in terms of the extension of the hazard zone with regard to the use of a 10 m resolution DEM is addressed through our response to Comment 2.6 of Reviewer 1 (see Section 4.3 and Supplement 7).

[Figure]

**Figure R2**. View of water depth map between cross-sections 1 and 2, corresponding to the breaching-scenario, with a 50 year-flood.

[revised manuscript text omitted]

---

## Referee Report (RR1)

885

[referee-annotated manuscript omitted]